# MIDUS: MEMORY-INFUSED DEPTH UP-SCALING

## ABSTRACT

Scaling large language models (LLMs) demands approaches that increase capacity without incurring excessive parameter growth or inference cost. Depth Up-Scaling (DUS) has emerged as a promising strategy by duplicating layers and applying Continual Pre-training (CPT), but its reliance on feed-forward networks (FFNs) limits efficiency and attainable gains. We introduce Memory-Infused Depth Up-Scaling (MIDUS), which replaces FFNs in duplicated blocks with a head-wise memory (HML) layer. Motivated by observations that attention heads have distinct roles both across and within layers, MIDUS assigns an independent memory bank to each head, enabling head-wise retrieval and injecting information into subsequent layers while preserving head-wise functional structure. This design combines sparse memory access with head-wise representations and incorporates an efficient per-head value factorization module, thereby relaxing the usual efficiency–performance trade-off. Across our CPT experiments, MIDUS exhibits robust performance improvements over strong DUS baselines while maintaining a highly efficient parameter footprint. Our findings establish MIDUS as a compelling and resource-efficient alternative to conventional FFN replication for depth up-scaling by leveraging its head-wise memory design.[1]

## 1 INTRODUCTION

Large language models (LLMs) improve predictably as parameters and data scale, yet training ever larger models from scratch is increasingly impractical in time and memory. A pragmatic alternative is model expansion, where a strong pre-trained backbone is enlarged and then further pre-trained. Depth Up-Scaling (DUS) expands a model by inserting new Transformer blocks either at the top of the stack or interleaved within it, or by selectively replicating only a subset of layers (Kim & Jung, 2020; Gong et al., 2019; Yang et al., 2020; Wu et al., 2024). DUS reuses the representations learned from the base model to accelerate convergence and stabilize optimization (Pan et al., 2024; Yano et al., 2025). Unlike width-oriented Mixture-of-Experts (MoE) that introduce sparsely routed experts (Shazeer et al., 2017),

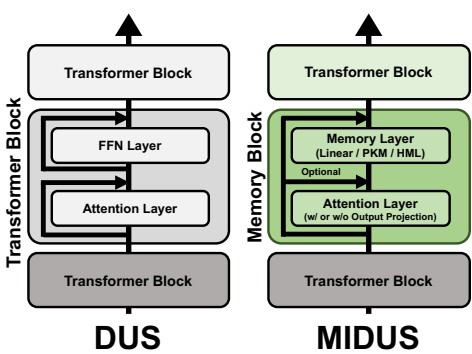

Figure 1: DUS upscales by depth-wise insertion of Transformer blocks, whereas MIDUS upscales by inserting Memory blocks.

DUS preserves a single dense path and integrates cleanly with standard training and inference stacks. Increasing depth also boosts representational power within a dense topology, which is known to be expressive for a given parameter budget (Rolnick & Tegmark, 2017; Liang & Srikant, 2016).

Recent work on DUS refines how depth is added rather than changing the underlying computation. Some methods decide where duplicated layers are inserted or which subset of layers is replicated (Wu et al., 2024; Kim et al., 2023), and others modify initialization and training strategies to stabilize optimization as depth grows (Yang et al., 2025; Cao et al., 2025). More recent variants introduce routing modules that choose, for each token, which layers participate in computation to bypass or sparsify unused capacity (Raposo et al., 2024; Tan et al., 2024). However, all of these methods still expand capacity by duplicating full Transformer blocks in which the FFN dominates both parameter

---

[1]Code: `https://anonymous.4open.science/r/iclr2026_submission_3045`

count and activation footprint. Each added block substantially increases model size, training memory, and inference latency, and dense FFNs remain resident on GPU even when routing is sparse. This tight coupling between extra capacity and FFN replication creates a structural bottleneck and motivates designs that add capacity without carrying full FFN pathways in every added block.

We start from the observation that capacity is not used uniformly. Layers specialize in different kinds of knowledge, and even within a layer, attention heads learn distinct functions whose importance shifts with context and task (Fernandez et al., 2024; Zhu et al., 2025; Yin & Steinhardt, 2025). Conventional DUS treats this heterogeneity as if it were homogeneous, with duplicated blocks applying a single dense FFN to the head-concatenated representation and offering no mechanism to channel extra capacity toward the heads that need it most. This work introduces Memory-Infused Depth Up-Scaling (MIDUS), a drop-in replacement for FFN-based Transformer block expansion that scales depth through retrieval rather than dense projection. Concretely, we replace selected Transformer blocks with Memory blocks, each consisting of a key–value memory layer, and interleave them following DUS policy. With identity-preserving initialization of the value table, the expanded model reproduces the base model's outputs at the start of training, ensuring stability. As added capacity is carried by retrieval, not by large dense matrices, MIDUS decouples quality gains from the parameter and activation costs of FFN.

To target capacity where it matters, we introduce the Head-wise Memory Layer (HML), which aligns memory retrieval with the attention mechanism via a multi-head attention without the output projection. Concretely, the embedding of each head serves directly as its query, eliminating the need for a separate query projection. Since HML assigns an independent memory bank to each head, it requires a design that can store and manage these per-head memories efficiently. We address this by pairing each attention head one-to-one with a Product-Key Memory (PKM) head (Lample et al., 2019). For each head, "row" and "column" sub-keys support a two-dimensional product-key lookup over the Cartesian product of sub-key choices, while keeping the lookup cost small. This design reduces both computation and key parameters compared to flat memories. To scale values efficiently, we introduce Head-wise Implicit Value Expansion (HIVE), which allocates per-head value capacity by keeping a single base value bank of head width and applying small head-specific transforms to produce head-width outputs that are concatenated back to model width, preserving head structure without storing separate value banks.

Empirically, MIDUS–HML achieves stronger language and reasoning performance than strong DUS baselines across general-purpose and math-domain benchmarks on Llama-3.2-1B and Llama-3.1-8B, while using fewer parameters, less GPU memory, and lower computational cost, and these gains persist under both CPT and SFT. Our contributions are summarized as follows. (1) We propose MIDUS, a memory-based alternative to FFN replication for DUS that interleaves Memory blocks into pre-trained LLMs. (2) We efficiently allocate per-head memory banks via HML and HIVE, aligning memory with attention heads and scaling key–value capacity effectively. (3) We validate MIDUS–HML through extensive experiments on Llama-3.2-1B and Llama-3.1-8B under CPT and SFT, across general-purpose and math-domain benchmarks, together with ablation studies on memory capacity, layer design, depth allocation, head importance, and efficiency, supporting MIDUS with head-wise memory as a practical and scalable direction for DUS.

## 2 RELATED WORKS

### 2.1 DEPTH UP-SCALING

A canonical path to model up-scaling is width expansion via MoE, where tokens are routed to a small subset of expert FFNs, boosting effective capacity without proportional dense compute (Shazeer et al., 2017; Jiang et al., 2024). Complementary to width, DUS enlarges models by duplicating or reorganizing Transformer blocks. Early stacking and progressive curricula study which layers to copy and how to schedule growth (Gong et al., 2019; Yang et al., 2020; Du et al., 2024; Saunshi et al., 2024; Pan et al., 2024; Yano et al., 2025). In decoder-only LLMs, Llama Pro scales by selectively copying a subset of layers under a stabilizing training recipe (Wu et al., 2024), while SOLAR deepens by stacking contiguous groups of layers in one step with a curated protocol (Kim et al., 2023). Recent lines emphasize initialization of the duplicates rather than placement. LESA learns a per-layer initializer for inserted blocks (Yang et al., 2025). OpT-DeUS proposes an optimal-

transport–based initializer with Avg-DeUS as a baseline, where the new layer is initialized by averaging the weights of its adjacent Transformer blocks (Cao et al., 2025).

DUS is attractive in practice, but they primarily add capacity through expanded FFNs, causing compute, activation memory, and latency to grow roughly linearly with the number of replicated blocks. Some research alleviates this dependence by introducing a routing module, analogous to MoE, that selects, for each token, which layers participate in computation (Raposo et al., 2024; Tan et al., 2024). However, all blocks still need to be kept active as candidates during inference, and performance remains tightly constrained by the computational budget. In contrast, MIDUS replaces the FFNs in selected Transformer blocks with HML, attaching additional capacity directly at the attention-head level. This design substantially reduces the total number of parameters and computation cost required for up-scaling while aligning capacity with head specialization and enabling on-demand retrieval, thereby yielding a more favorable capacity–efficiency trade-off for DUS.

## 2.2 Memory Layers and External Memory Architectures

A straightforward memory layer associates each slot with a flat key and value, accessed via dot-product lookup. However, such Linear memories exhibit poor scalability, since both lookup cost and key parameters grow linearly with memory size, making large key spaces computationally expensive. PKM (Lample et al., 2019) replaces part of this dense expansion with sparse retrieval over a key space formed as the Cartesian product of two codebooks. A two-stage Top-$k$ over $n$ row and $n$ column sub-keys addresses $n^2$ composite keys while evaluating only a small number of dot products, improving both compute and parameter efficiency compared to Linear memory. PKM can be used alongside or instead of FFNs, allowing tokens to consult few values rather than always traversing wide dense layers (Kim & Jung, 2020). Ultra-sparse memory systems further manage large banks via implicit value expansion and compressed representations, decoupling logical capacity from physical storage and keeping per-token compute modest (Huang et al., 2024; 2025).

HML adopts efficient PKM-based key lookup (Lample et al., 2019) and an implicit-value approach (Huang et al., 2024; 2025), but makes the memory head-wise rather than block-shared. Each attention head owns its own key space while values are produced from a shared value bank via lightweight head-specific projections. This design respects the distinct roles of individual heads and enables selective retrieval tailored to each head, without a one-to-one increase in stored values. Together, the introduction of head-wise memory banks and efficient memory management keeps retrieval overhead low and provides an effective alternative to duplicating dense FFNs for DUS.

## 2.3 Roles of Heads in Transformer Layers

Empirical studies consistently show that attention heads are specialized and vary in importance across layers and tasks. Gradient- and ablation-based analyses reveal layer-dependent cues (Fernandez et al., 2024), attribution finds that a small subset of heads contributes disproportionately while others remain quiescent (Zhu et al., 2025), and distributional evidence indicates distinct role profiles (for example, local syntax versus long-range semantics) (Yin & Steinhardt, 2025). Treating this heterogeneous structure as homogeneous through uniform, Transformer block-level augmentation risks misallocating capacity and reinforcing inactive pathways. Motivated by these observations, MIDUS allocates memory at the granularity of heads, equipping each head with a memory bank that stores and retrieves information it uniquely exploits, thereby reducing interference from block-shared memories and focusing capacity where it is most useful.

## 3 Preliminaries

We establish the notation used throughout. Specifically, we formalize (i) the standard pre-norm Transformer block with residual connections and (ii) a generic linear memory layer. These two primitives will serve as the basic components from which MIDUS is defined in the next section.

**Transformer block.** Let $x \in \mathbb{R}^{s \times d}$ be the input with $s$ tokens and hidden dimension $d$. A pre-norm Transformer block with residual connections is

$$a = x + \text{Attn}\big(\text{LN}_{\text{Attn}}(x)\big), \qquad y = a + \text{FFN}\big(\text{LN}_{\text{FFN}}(a)\big). \tag{1}$$

We refer to the map realized above as the *Transformer block*

$$T : \mathbb{R}^{s \times d} \to \mathbb{R}^{s \times d}, \qquad T(x) = y, \tag{2}$$

and use $T_\ell$ to denote the $\ell$-th block.

**Linear Memory layer.** We denote a memory layer by

$$\text{Mem} : \mathbb{R}^{s \times d} \to \mathbb{R}^{s \times d}, \qquad \text{Mem}(a) = m, \tag{3}$$

where $a, m \in \mathbb{R}^{s \times d}$. A memory layer is specified by a query map, a key–value memory bank. A representative instance of a memory layer is the Linear, or flat, memory layer. Define the query $q(a) = aW_q \in \mathbb{R}^{s \times d_q}$ with learnable $W_q \in \mathbb{R}^{d \times d_q}$, keys $K \in \mathbb{R}^{N \times d_q}$, and values $V \in \mathbb{R}^{N \times d}$ for $N$ memory slots. The similarity scores and indices are

$$S(a) \;=\; q(a)\,K^\top \;\in\; \mathbb{R}^{s \times N}, \qquad \Omega_r \;=\; \text{Top}_k\big(S(a)_r\big) \in \{1, \ldots, N\}^k \tag{4}$$

for each token $r \in \{1, \ldots, s\}$. Here, $S(a)_r$ denotes the $r$-th row of $S(a)$ (the similarity between token $r$ and all memory slots), and $S(a)_{r,\Omega_r}$ denotes the entries of $S(a)_r$ indexed by $\Omega_r$. The set of normalized weights is $\alpha_r \in \Delta^{|\Omega_r|}$, obtained by applying a softmax to $S(a)_{r,\Omega_r}$. From the per-token weighted sum of selected values,

$$m_r \;=\; \sum_{j \in \Omega_r} \alpha_{r,j}\, V_j \;\in\; \mathbb{R}^d, \qquad m \;=\; \big[m_1; \ldots; m_s\big] \in \mathbb{R}^{s \times d}. \tag{5}$$

## 4 METHODOLOGY

We present MIDUS, a memory-based alternative to FFN replication for DUS. A Memory block replaces the FFN in a Transformer block with a key–value memory layer, and such blocks can be interleaved to realize DUS in a drop-in manner. We also describe initialization and the residual path preserving the base model at training start. Finally, we introduce HML, a per-head memory design built on product-key lookup and efficient value factorization.

### 4.1 MEMORY-INFUSED DEPTH UP-SCALING

**Memory block.** We define the Memory block $M$ by replacing the FFN of a Transformer block with a memory layer:

$$a \;=\; x \;+\; \text{Attn}\big(\text{LN}_{\text{Attn}}(x)\big), \qquad m \;=\; \text{Mem}(a). \tag{6}$$

With a residual connection, the block output is

$$M : \mathbb{R}^{s \times d} \to \mathbb{R}^{s \times d}, \qquad M(x) = x + m. \tag{7}$$

The attention layer first contextualizes tokens across the sequence, and the memory layer then retrieves information by querying a key–value bank and aggregating selected values. Lample et al. (2019) report that substituting a small number of FFNs with memory layers improves pre-training performance. We build on this idea and use Memory blocks as the vehicle for scaling depth. Added depth is carried by retrieval-based capacity rather than dense feed-forward projections.

**Memory-Infused Depth Up-Scaling.** Let the base model be the composition of $L$ Transformer blocks

$$f_0(x) \;=\; T_{L-1} \circ \cdots \circ T_0(x). \tag{8}$$

A representative strategy in DUS is to *interleave* $K$ additional Transformer blocks $\widetilde{T}_0, \ldots, \widetilde{T}_{K-1}$ at specified layer indices $0 \le i_0 < \cdots < i_{K-1} \le L - 1$ (Wu et al., 2024). The DUS model is

$$f_{\text{DUS}}(x) \;=\; T_{L-1} \circ \cdots \circ \widetilde{T}_{K-1} \circ T_{i_{K-1}} \circ \cdots \circ \widetilde{T}_0 \circ T_{i_0} \circ \cdots \circ T_0(x). \tag{9}$$

MIDUS follows the interleaving rule, replacing $\widetilde{T}_{0:K-1}$ with Memory blocks $M_0, \ldots, M_{K-1}$:

$$f_{\text{MIDUS}}(x) \;=\; T_{L-1} \circ \cdots \circ T_{i_{K-1}} \circ M_{K-1} \circ \cdots \circ T_{i_0} \circ M_0 \circ \cdots \circ T_0(x). \tag{10}$$

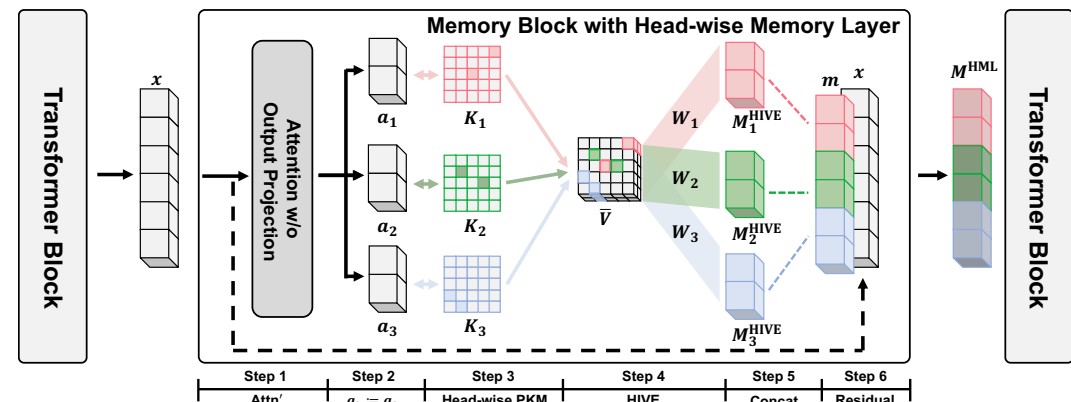

Figure 2: MIDUS with HML is divided into six steps. In this figure, we assume the case where the number of attention heads is $H = 3$ and $k = 2$ for Top-$k$ selection. Step 1) The input $x$ entering the Memory block passes through $\mathrm{Attn}'$, an attention layer without output projection, producing $a'$. Step 2) $a'$ is split into head-wise representations $a_h$, which directly serve as the queries $q_h$ for each head. Step 3) Each head is associated with a product-key set $K_h$, and the $k$ most similar keys are retrieved for $q_h$. Step 4) From the shared value set $\bar{V}$, the corresponding values for each head are found and aggregated with weighted sums, which are then projected by the head-specific matrices $W_h$. Step 5) The head-wise memory outputs $M_h^{\mathrm{HIVE}}$ are concatenated to form $m$. Step 6) Finally, $m$ is added to the input $x$ via a residual connection, realizing head-wise memory infusion.

The ordering difference between Eq.9 and Eq.10 is deliberate and distinguishes MIDUS from designs of Wu et al. (2024). Placing each Memory block after its corresponding Transformer block allows it to access the contextual representation that will be processed by the next layer, enabling the model to exploit subsequent context more directly. This realizes extra depth while decoupling the added capacity from dense FFNs. In this work, MIDUS does not address the case of replacing the stacked Transformer blocks used in DUS (Kim et al., 2023) with stacked Memory blocks. Figure 1 illustrates the structural differences between DUS and MIDUS.

**Weight initialization and residual connection.** Recall from Eq.7 that a Memory block outputs is $M(x) = x + m$, where $m = \mathrm{Mem}(a)$ is constructed from values retrieved from the memory bank. We initialize these values to zero so that $\mathrm{Mem}(a) = 0$ at the start of training. Under this initialization, the Memory block reduces to the identity map $M(x) = x$, ensuring that the expanded model initially produces the same output as the base model. This preserves the base model's performance in the early training phase and provides a stable starting point for learning memory parameters.

In addition, the attention layer within each Memory block is initialized with the same parameters as the attention layer of the subsequent pre-trained Transformer block. At initialization, this design ensures that retrieval from the memory bank is conditioned on the same contextual information that the next Transformer block already uses. The retrieved values are then injected into the input of that subsequent block through the residual connection in Eq.7, allowing newly learned knowledge to be smoothly infused into the base model's representations without disrupting its original capacity.

## 4.2 HEAD-WISE MEMORY LAYER

Inspired by evidence that different attention heads specialize in distinct functions (Fernandez et al., 2024; Zhu et al., 2025; Yin & Steinhardt, 2025), we posit that the information most useful for retrieval also varies by head. We assign a separate memory bank to each attention head and generate head-specific queries, so that each head stores and retrieves patterns adapted to its role. To make per-head retrieval both fast and scalable, we adopt a multi-head product-key design.

**Multi-head product-key memory.** We introduce memory heads partitioning the memory layer into $H$ parallel query–key lookups aligned with $H$ attention heads. Classical PKM (Lample et al., 2019) also uses multiple memory heads. Our design keeps these mechanics and relies on row–column key factorization, additive pair scoring, and two-stage Top-$k$ selection. We set the number of memory heads to $H$ so that each attention head pairs with a memory head.

Starting from $q(a) = aW_q \in \mathbb{R}^{s \times d_q}$, form head-wise slices $q_h \in \mathbb{R}^{s \times 2d_p}$ with $d_p = \frac{d_q}{2H}$ and split $q_h = [\, q_h^{\text{row}} \mid q_h^{\text{col}} \,]$, where $q_h^{\text{row}}, q_h^{\text{col}} \in \mathbb{R}^{s \times d_p}$. For each head $h$, keep sub-key banks $K_h^{\text{row}}, K_h^{\text{col}} \in \mathbb{R}^{n \times d_p}$ whose Cartesian product yields $N = n^2$ composite keys indexed by $\pi(i,j) = (i-1)n + j$. For token index $r \in \{1, \ldots, s\}$, compute scores

$$S_{h,r}^{\text{row}} = q_{h,r}^{\text{row}} K_h^{\text{row}\top} \in \mathbb{R}^n, \qquad S_{h,r}^{\text{col}} = q_{h,r}^{\text{col}} K_h^{\text{col}\top} \in \mathbb{R}^n. \tag{11}$$

Select top-$k$ row and column index sets

$$I_{h,r} = \text{Top}_k\big(S_{h,r}^{\text{row}}\big) \in \{1, \ldots, n\}^k, \qquad J_{h,r} = \text{Top}_k\big(S_{h,r}^{\text{col}}\big) \in \{1, \ldots, n\}^k, \tag{12}$$

form the $k^2$ candidate pairs $\Omega_{h,r} = I_{h,r} \times J_{h,r}$, and define additive pair scores

$$\sigma_{h,r}(i,j) = S_{h,r}^{\text{row}}(i) + S_{h,r}^{\text{col}}(j), \qquad (i,j) \in \Omega_{h,r}. \tag{13}$$

Apply a second $\text{Top}_k$ over $\{\sigma_{h,r}(i,j)\}_{(i,j) \in \Omega_{h,r}}$ to obtain the final pair set $\widehat{\Omega}_{h,r}$ and normalize the corresponding scores over $\widehat{\Omega}_{h,r}$ to get weights $\alpha_{h,r}$. With $V \in \mathbb{R}^{N \times d}$,

$$M_{h,r}(a) = \sum_{(i,j) \in \widehat{\Omega}_{h,r}} \alpha_{h,r}(i,j) \, V_{\pi(i,j)} \in \mathbb{R}^d. \tag{14}$$

Stacking over $r$ gives $M_h(a) \in \mathbb{R}^{s \times d}$, and the memory output aggregates heads

$$m = \sum_{h=1}^{H} M_h(a) \in \mathbb{R}^{s \times d}. \tag{15}$$

Compared with a multi-head linear memory, multi-head PKM keeps $N$ addressable keys while reducing compute and key parameters. One $N$-way scores–and–select over $H$ heads costs $\mathcal{O}(s\,N\,d_q)$, whereas PKM replaces it with two $n$-way products per head using slices of width $d_p = \frac{d_q}{2H}$, giving $\mathcal{O}(s\,n\,d_q)$ in total, a $\sqrt{N}$ reduction since $N = n^2$. Key parameters drop from $Nd_q$ to $nd_q$ because each head stores only $2n$ sub-keys of width $d_p$.

However, multi-head PKM still shares value table $V \in \mathbb{R}^{N \times d}$ across heads, limiting head-wise specialization as heads with different roles retrieve from the same pool. Inspired by (Huang et al., 2024), we introduce Head-wise Implicit Value Expansion (HIVE), a novel scheme that allocates per-head value spaces to complete head-wise memory bank assignment.

**Head-wise Implicit Value Expansion.** We replace the width-$d$ value table with a head-aligned factorization matching the attention head. Let $d_h = d/H$, use base table $\bar{V} \in \mathbb{R}^{N \times d_h}$, and introduce per-head transforms $W_h \in \mathbb{R}^{d_h \times d_h}$. Reusing $\widehat{\Omega}_{h,r}$ and $\alpha_{h,r}$, per-token, per-head aggregation is

$$\bar{M}_{h,r} = \sum_{(i,j) \in \widehat{\Omega}_{h,r}} \alpha_{h,r}(i,j) \, \bar{V}_{\pi(i,j)} \in \mathbb{R}^{d_h}, \qquad M_{h,r}^{\text{HIVE}} = W_h \bar{M}_{h,r} \in \mathbb{R}^{d_h}. \tag{16}$$

Stacking over $r$ yields $M_h^{\text{HIVE}} \in \mathbb{R}^{s \times d_h}$, and the memory output concatenates heads

$$m = \big[\, M_1^{\text{HIVE}} \mid M_2^{\text{HIVE}} \mid \cdots \mid M_H^{\text{HIVE}} \,\big] \in \mathbb{R}^{s \times d}. \tag{17}$$

This implicitly allocates value spaces per head without storing $H$ separate value banks. A naïve head-wise table uses $HNd_h$ parameters, whereas the factorized form uses $Nd_h + Hd_h^2$. Computational overhead remains modest, since the PKM lookup is unchanged and each head applies only a $d_h \times d_h$ map once per token. By producing head-width outputs and concatenating them, the retrieved information preserves head structure and is naturally injected into the next Transformer block.

**Head-wise Memory Layer.** We align memory queries with attention heads and remove the explicit query projection, and omit the residual connection across the attention layer. Let

$$a' = \text{Attn}'(\text{LN}_{\text{Attn}}(x)) = \big[\, a_1 \mid a_2 \mid \cdots \mid a_H \,\big] \in \mathbb{R}^{s \times d}, \tag{18}$$

where $a_h \in \mathbb{R}^{s \times d_h}$ and $d_h = d/H$. Here $\text{Attn}'$ is multi-head attention *without the output projection*, so the output is the concatenation of per-head embeddings, and we take the head embeddings

Table 1: Comparison of CPT and SFT on Llama-3.2-1B across DUS baselines and MIDUS.

| | Method | Perplexity ↓ Wiki-PPL | Zero-shot Accuracy ↑ ARC | LogiQA | Wino | CSQA | BoolQ | PIQA | MMLU | Average |
|---|---|---|---|---|---|---|---|---|---|---|
| **CPT-1B** | Base | 13.22 | 68.69 | 22.27 | 60.22 | 25.88 | 63.61 | 75.08 | 29.85 | 49.37 |
| | SOLAR | 13.41 | **68.81** | 22.58 | 60.22 | 26.29 | 61.16 | **75.24** | 30.14 | 49.21 |
| | Llama Pro | 12.33 | 66.75 | 22.43 | 59.75 | 35.14 | 64.86 | 74.54 | 31.22 | 50.67 |
| | LESA | 12.06 | 66.33 | 21.97 | 60.22 | 45.21 | 64.37 | 74.59 | 36.11 | 52.69 |
| | OpT-DeUS | 11.72 | 66.46 | **23.35** | **61.56** | 46.44 | 62.42 | 74.54 | 36.56 | 53.05 |
| | Avg-DeUS | 12.04 | 66.84 | 22.73 | 60.38 | 41.93 | 64.01 | 74.27 | 34.40 | 52.08 |
| | MIDUS-HML | **11.64** | 66.16 | 23.20 | **61.56** | 46.27 | **65.29** | 75.08 | **36.91** | **53.50** |
| **SFT-1B** | Base | 13.09 | **69.65** | 21.20 | 58.96 | 26.37 | 62.66 | 75.46 | 30.43 | 49.25 |
| | SOLAR | 13.23 | 69.28 | **23.81** | 58.33 | 27.44 | 60.64 | **75.90** | 31.05 | 49.49 |
| | Llama Pro | 12.55 | 67.68 | 22.89 | 60.06 | 42.59 | 62.75 | 75.46 | 33.62 | 52.15 |
| | LESA | 12.37 | 65.99 | 23.35 | 60.38 | 47.91 | **66.33** | 75.24 | 36.43 | 53.66 |
| | OpT-DeUS | 11.82 | 66.62 | **23.81** | 60.14 | 49.55 | 63.43 | 75.52 | 37.64 | 53.81 |
| | Avg-DeUS | 12.18 | 67.09 | 23.66 | 60.30 | 47.01 | 65.63 | 74.97 | 35.62 | 53.47 |
| | MIDUS-HML | **11.75** | 66.33 | 21.51 | **60.77** | **50.04** | 66.21 | **75.90** | **37.82** | **54.08** |

themselves as queries, $q_h := a_h$. We denote by $\mathrm{HML} : \mathbb{R}^{s \times d} \to \mathbb{R}^{s \times d}$ the head-wise memory layer, implemented without query projection via per-head PKM selection and HIVE aggregation. The resulting memory block is

$$M^{\mathrm{HML}}(x) = x + \mathrm{HML}\big(\mathrm{Attn}'(\mathrm{LN}_{\mathrm{Attn}}(x))\big). \tag{19}$$

Consequently, in MIDUS we instantiate Memory blocks replacing Transformer blocks as HML blocks $M^{\mathrm{HML}}$. Figure 2 illustrates the detailed step-by-step operation of MIDUS with HML.

# 5 EXPERIMENTS

## 5.1 EXPERIMENTS SETTINGS

Following (Cao et al., 2025), our base models are Llama-3.2-1B and Llama-3.1-8B (Dubey et al., 2024). We add 8 and 16 additional blocks to these models, respectively, for each DUS and MIDUS variant. For MIDUS, unless otherwise specified, each memory layer is implemented with a product-key bank with $n = 64$ sub-keys per row/column, i.e., $N = n^2 = 4096$ composite keys per head, and Top-$k$ selection with $k = 4$. Both base models have 32 attention heads per block. Consequently, for Llama-3.2-1B, replacing 8 blocks with HML memory blocks yields $32 \times 64^2 \times 8 \approx 1M$ total memory slots, and by the same calculation, Llama-3.1-8B utilizes approximately 2M slots. Throughout the result tables, boldface denotes the best score, and underlining denotes the second-best. Further experimental details, hyperparameter settings, and ablation studies are provided in the Appendix.

**DUS placement policy.** Block placement depends on the DUS method. Llama Pro (Wu et al., 2024) employs a uniform depth-wise policy, with blocks evenly inserted and initialized from the preceding block. MIDUS follows the same policy but initializes each added block from the subsequent block. We call this the *Distributed* policy, consistent with the difference between Eq.9 and Eq.10. Alternatively, added blocks may be concentrated in upper blocks (*Top-heavy*), as in OpT-DeUS (Cao et al., 2025) and LESA (Yang et al., 2025), or in lower blocks (*Bottom-heavy*). Up-scaled Llama-3.2-1B has 24 layers, and the added-block indices for each policy are as follows:

- *Top-heavy*: {8, 10, 12, 14, 16, 18, 20, 22}
- *Distributed*: {1, 4, 7, 10, 13, 16, 19, 22}
- *Bottom-heavy*: {0, 2, 4, 6, 8, 10, 12, 14}

Unless otherwise noted, MIDUS variants follow the *Distributed* policy. Refer to Appendix D for details and Table 11 for the ablation studies on different DUS policies.

**Continual Pre-training.** We perform CPT in two settings: one on the 1.5B-token `FineWeb-Edu` subset (Penedo et al., 2024; Cao et al., 2025) to target general-purpose language and reasoning, and another on a 1.1B-token `MathPile` subset (Wang et al., 2024) to specialize the model on math-domain abilities. For CPT, we train only the newly inserted blocks and freeze all pre-existing base blocks (Wu et al., 2024; Yang et al., 2025; Cao et al., 2025), whereas for the base model and SOLAR (Kim et al., 2023), all parameters are updated during training.

Table 2: Comparison of CPT and SFT on Llama-3.1-8B across DUS baselines and MIDUS. Entries marked with † are taken from (Cao et al., 2025).

| | Method | Perplexity ↓ | Zero-shot Performance ↑ | | | | | | | |
|---|---|---|---|---|---|---|---|---|---|---|
| | | Wiki-PPL | ARC | LogiQA | Wino | CSQA | BoolQ | PIQA | MMLU | Average |
| **CPT-8B** | Base† | 8.35 | 79.97 | 26.88 | 72.06 | 65.19 | 81.83 | 78.84 | 58.61 | 66.20 |
| | SOLAR† | 9.90 | 79.88 | 26.88 | 71.59 | 57.41 | 80.70 | 78.56 | 54.37 | 64.20 |
| | Llama Pro† | 7.81 | 81.61 | **29.49** | 73.72 | 70.93 | 81.65 | 79.98 | 62.56 | 68.56 |
| | LESA† | 7.73 | 82.07 | 27.96 | 74.11 | **72.40** | 81.93 | 80.30 | 62.63 | 68.77 |
| | OpT-DeUS† | 7.73 | 82.07 | 27.34 | **74.74** | 71.91 | 82.26 | **80.79** | 62.96 | 68.87 |
| | Avg-DeUS† | 7.95 | 82.15 | 27.50 | 73.48 | 71.09 | 82.17 | 80.20 | 62.11 | 68.39 |
| | MIDUS-HML | **7.40** | **82.37** | 28.57 | 74.59 | 70.84 | **82.87** | 80.25 | **63.40** | **68.98** |
| **SFT-8B** | Base† | 8.32 | 81.10 | 24.58 | 72.14 | 68.30 | 82.14 | 79.71 | 59.17 | 66.73 |
| | SOLAR† | 9.68 | 80.68 | 25.19 | 71.19 | 61.81 | 81.19 | 79.16 | 55.03 | 64.80 |
| | Llama Pro† | 7.81 | 83.33 | 27.19 | 74.11 | 72.07 | 82.26 | 80.79 | 62.32 | 68.87 |
| | LESA† | 7.72 | 83.84 | 26.57 | 75.53 | 73.05 | 83.00 | 80.69 | 63.57 | 69.47 |
| | OpT-DeUS† | 7.73 | 83.80 | 26.73 | **76.09** | 73.05 | 83.36 | 80.85 | 63.84 | 69.67 |
| | Avg-DeUS† | 7.91 | **83.88** | 26.42 | 75.45 | 72.89 | 83.18 | 80.47 | 63.10 | 69.34 |
| | MIDUS-HML | **7.50** | 83.50 | **28.11** | 74.90 | **73.63** | **83.39** | **80.96** | **64.54** | **69.86** |

**Supervised Fine-tuning.** We further apply supervised fine-tuning (SFT) to the CPT model trained on the `FineWeb-Edu` subset, fine-tuning it separately on two SFT datasets, `Alpaca-GPT4` (Peng et al., 2023) and `Databricks-Dolly-15k` (Conover et al., 2023), to evaluate how CPT-induced improvements transfer to instruction-following tasks. During SFT, all parameters are updated for every method, following previous studies (Yang et al., 2025; Cao et al., 2025).

**Benchmarks.** We evaluate models using the `lm-evaluation-harness` framework (Gao et al., 2021). For general-purpose language and reasoning, we adopt the knowledge-centric suite used by Cao et al. (2025): ARC-Easy (Clark et al., 2018), LogiQA (Liu et al., 2020), Winogrande (Sakaguchi et al., 2021), CSQA (Talmor et al., 2018), BoolQ (Clark et al., 2019), PIQA (Bisk et al., 2020), and MMLU (Hendrycks et al., 2020), along with perplexity on WikiText (Merity et al., 2016). To assess math-domain reasoning, we report performance on GSM8K, GSM8K–CoT (Cobbe et al., 2021), MATH (Hendrycks et al., 2020), and MathQA (Amini et al., 2019). General-purpose benchmarks adopt zero-shot evaluation, whereas math-domain benchmarks utilize 5-shot evaluation.

## 5.2 RESULTS

**CPT and SFT results on `FineWeb-Edu`.** Table 1 and Table 2 report general-purpose language and reasoning performance for models CPT-trained on the `FineWeb-Edu` subset with Llama-3.2-1B and Llama-3.1-8B backbones, respectively. Each table includes the base model without any up-scaling (Base), several DUS methods, and MIDUS with HML memory layers (MIDUS-HML), evaluated both before and after additional SFT on `Alpaca-GPT4`. Across both backbones, MIDUS-HML attains the lowest perplexity on Wiki-PPL and the highest average zero-shot accuracy. This indicates that MIDUS-HML is a more effective up-scaling strategy than DUS, since replacing FFNs with head-wise memory layers allows different heads to specialize and store complementary knowledge. Low perplexity and strong average performance also persist after SFT, suggesting that knowledge stored in the memory bank during CPT is successfully transferred through SFT into instruction-following ability that rivals or surpasses that of conventional Transformer blocks. A similar pattern appears under SFT on `Databricks-Dolly-15k`, as reported in Table 6.

Table 3 reports CPT performance of MIDUS with different memory layers. For a fair comparison, all memory layer designs ensure that each attention head retrieves from a head-wise distinct memory. That is, not only HML but also the Linear and PKM layers maintain a single key bank per head, and the attention output is split across heads and used as queries. The Linear layer performs a flat query–key lookup (Eq.4), whereas PKM adopts a product-key design (Eq.11). Following prior work (Lample et al., 2019), queries in these two memory layers are batch-normalized, and their value memory bank is shared across all heads. In contrast, HML treats the $\text{Attn}'$ output directly as the query without any normalization and, by leveraging HIVE, allocates both key and value memory banks in a head-wise manner. Table 3 implies that HML substantially outperforms the Linear and PKM memory layers, suggesting that simply introducing conventional memory layers in MIDUS is insufficient to guarantee strong up-scaled performance, and that introducing head-wise key–value

Table 3: CPT results on `FineWeb-Edu` with Llama-3.2-1B for MIDUS variants with different memory layers. Linear and PKM use the same number of memory heads as HML. Unlike HML, only the key memories are head-wise, while the value memory bank remains shared across heads.

| | Perplexity ↓ | Zero-shot Accuracy ↑ | | | | | | | |
|---|---|---|---|---|---|---|---|---|---|
| Memory Layer | Wiki-PPL | ARC | LogiQA | Wino | CSQA | BoolQ | PIQA | MMLU | Average |
| Linear | 11.82 | 65.61 | 22.73 | **62.04** | 42.42 | 64.71 | **75.08** | 35.89 | 52.64 |
| PKM | 11.97 | 65.70 | 21.35 | 61.40 | 42.18 | 65.47 | 74.65 | 33.93 | 52.10 |
| HML | **11.64** | **66.16** | **23.20** | 61.56 | **46.27** | 65.29 | **75.08** | **36.91** | **53.50** |

Table 4: Efficiency of DUS and MIDUS for Llama-3.1-8B, with the number of parameters (trainable/total), GPU memory (train/inference), training time, and generation throughput.

| | Methods | B ↓ Params | GB ↓ GPU Memory | s/iter ↓ Train | Tokens/s ↑ Throughput |
|---|---|---|---|---|---|
| | Base | 8.03 / 8.03 | 81.6 / 19.2 | 6.12 | 61.0 |
| DUS | Llama Pro | 3.49 / 11.52 | 63.8 / 27.0 | 6.89 | 41.3 |
| DUS | OpT-DeUS | 3.49 / 11.52 | 59.8 / 27.0 | 5.78 | 41.3 |
| MIDUS | Linear | 1.48 / 9.51 | 51.3 / 23.1 | 11.05 | 42.4 |
| MIDUS | PKM | 1.21 / 9.24 | 48.3 / 22.5 | 9.88 | 47.3 |
| MIDUS | HML | **0.42 / 8.45** | **34.8 / 20.1** | 5.05 | 50.5 |

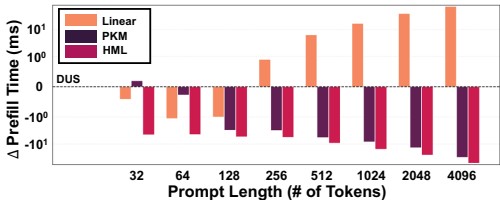

Figure 3: Change in prefill time versus prompt length for each memory layer relative to DUS, where negative $\Delta$ Prefill Time indicates faster prefill time than DUS on Llama-3.1-8B.

memory banks is crucial. The effect of normalization and the attention output projection within the HML architecture is further analyzed in Appendix F.4.

**Efficiency of MIDUS-HML.** Table 4 summarizes parameter count, peak GPU memory, training latency, and generation throughput for DUS and MIDUS variants on Llama-3.1-8B. Among upscaling methods, MIDUS-HML exhibits the strongest efficiency profile. It uses the fewest trainable and total parameters, achieves the lowest peak training memory, and outpaces the fastest DUS baseline, OpT-DeUS, in both training time and generation throughput. Figure 3 further analyzes prefill efficiency as a function of prompt length. Compared with Linear and PKM memory layers, MIDUS-HML achieves lower prefill time than DUS across a broad range of prompt lengths, including short prompts, with the largest gains at the longest prompts (negative $\Delta$ Prefill Time). Note that all DUS methods share identical inference-time efficiency and therefore correspond to a single DUS reference curve in Figure 3. The efficiency benefits of MIDUS-HML become even more pronounced on deeper and larger models. Additional efficiency results on the Llama-3.1-8B backbone are reported in Table 12 and Figure 5 in Appendix.

Independent of expanded-block structure, the DUS placement policy strongly affects efficiency (Cao et al., 2025). For example, OpT-DeUS concentrates trainable blocks near the top of the network (*Top-heavy*), so fewer layers participate in the backward pass and gradient computation is cheaper. By contrast, our default MIDUS configuration adopts the *Distributed* policy, placing trainable blocks deeper in the stack; gradients traverse more layers, making training slower despite the lighter memory layer. Nevertheless, MIDUS-HML remains comparable to, and often more efficient than, OpT-DeUS in most settings. When MIDUS-HML is configured with the same *Top-heavy* policy as OpT-DeUS, it consistently achieves higher training efficiency across batch sizes and sequence lengths, as shown in Figures 6 and 7. Although the DUS placement policy is important for MIDUS-HML, Table 11 shows that its performance remains comparable to DUS baselines under all considered policies. Appendix C describes the implementation-level optimizations already applied to MIDUS-HML, and further low-level improvements may yield additional efficiency gains.

**Amplification of head importance by HML.** We examine whether head-wise memory banks, which allow each head to manage and exploit distinct knowledge, actually amplify head-level contributions. To this end, we use the head importance score $IS_h$ proposed by Bansal et al. (2023). $IS_h$ quantifies the importance of head $h$ for a prompt input $x$ and answer $y$ based on the magnitude of its output $a_h([x|y])$ and the magnitude of the gradient of the loss $\mathcal{L}(y;x)$ with respect to that output,

$$IS_h(\mathcal{D}) := \mathbb{E}_{(x,y)} \left[ a_h([x|y])^T \frac{\partial \mathcal{L}(y;x)}{\partial a_h([x|y])} \right] \quad (20)$$

Here, $[x|y]$ denotes the sequence obtained by concatenating $x$ and $y$, and $\mathcal{D}$ denotes the task.

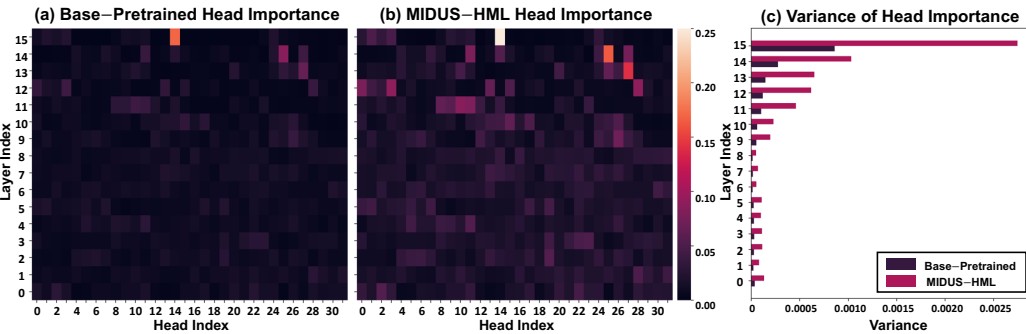

Figure 4: Head importance on PICA benchmark for Llama-3.2-1B. (a) Base-pretrained head importance. (b) MIDUS-HML head importance after CPT on `FineWeb-Edu`. (c) Per-layer variance of head importance, showing stronger and more concentrated heads under HML.

Table 5: CPT results on `MathPile` for Llama-3.2-1B and Llama-3.1-8B models.

| Method | CPT-1B: 5-shot Accuracy ↑ | | | | | CPT-8B: 5-shot Accuracy ↑ | | | | |
|---|---|---|---|---|---|---|---|---|---|---|
| | GSM8K | GSM8K–CoT | MATH | MathQA | Average | GSM8K | GSM8K–CoT | MATH | MathQA | Average |
| Base | 4.32 | 4.93 | 5.66 | 29.88 | 14.93 | 35.33 | 33.36 | 11.42 | 37.99 | 39.37 |
| Llama Pro | 5.61 | 4.40 | 5.64 | 30.62 | 15.42 | 41.93 | 41.77 | 13.10 | 40.87 | 45.89 |
| OpT-DeUS | 6.07 | 6.22 | **5.80** | 30.95 | 16.35 | **49.36** | 50.04 | 13.84 | **42.58** | 51.94 |
| MIDUS-HML | **6.60** | **6.60** | 5.62 | **31.39** | **16.73** | 48.90 | **51.78** | **14.34** | **42.58** | **52.53** |

We evaluate $IS_h$ on the PICA benchmark with zero-shot for two models: the base pretrained model before any up-scaling or CPT, and the MIDUS-HML model obtained by inserting HML memory layers and then performing CPT on `FineWeb-Edu`. For both models, we compute $IS_h$ for every head in each of the 16 Transformer layers. Figure 4 reports results for the Llama-3.2-1B backbone. The base model has 16 layers, and MIDUS-HML keeps all 16 base layers frozen, so their weights remain identical, although the presence of interleaved HML layers leads to different $IS_h$ across the two models. As shown in Figure 4-(a) and (b), HML increases head importance overall. In addition, heads that were already important for solving task $\mathcal{D}$ in the base model show larger gains in $IS_h$ than less important heads. Consequently, the variance of $IS_h$ across heads within each layer grows under HML, as illustrated in Figure 4-(c). These results indicate that the head-wise memory bank introduced by HML effectively strengthens and sharpens the functional roles of individual heads.

**CPT results on `MathPile`.** MIDUS-HML successfully learns not only general knowledge but also domain-specific knowledge that requires specialized and high-level reasoning. The `MathPile` dataset is constructed from diverse mathematical corpora (Wang et al., 2024), and we perform CPT on both Llama-3.2-1B and Llama-3.1-8B backbones. Table 5 reports 5-shot accuracy on math-domain reasoning benchmarks. Regardless of backbone size, MIDUS-HML consistently achieves the best results on almost all datasets. This suggests that the head-wise memory banks successfully store and appropriately exploit even difficult, high-level mathematical knowledge.

## 6 CONCLUSION

We introduce MIDUS, a novel depth up-scaling approach that departs from conventional DUS by replacing FFNs in Transformer blocks with memory layers. By introducing head-wise memory banks and designing the HML memory layer, MIDUS sharply reduces parameters, GPU memory, and computational cost while preserving depth-wise capacity. HML enables different attention heads to store and exploit distinct knowledge, allowing MIDUS–HML to learn both general and challenging mathematical knowledge and to outperform standard DUS across a range of backbones, SFT datasets, and benchmarks. Our head importance analysis further shows that HML amplifies and concentrates the roles of important heads. These results suggest that head-wise memory–based depth up-scaling is a promising alternative to traditional FFN replication for model up-scaling.

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

# A  APPENDIX

## A.1  STATEMENT ON THE USE OF LARGE LANGUAGE MODELS

LLMs were used exclusively for minor editorial tasks, including grammar correction and readability enhancement. All core scientific contributions—idea conception, methodological development, theoretical and experimental work, and manuscript drafting—were carried out entirely by the authors without LLM involvement.

# B  EXPERIMENTAL DETAILS

## B.1  HYPERPARAMETER SETTING

We conduct `FineWeb-Edu` experiments on NVIDIA RTX 6000 Blackwell GPUs and `MathPile` experiments on NVIDIA H200 GPUs. Across all experiments, we use the AdamW optimizer (Loshchilov & Hutter, 2017) with a cosine learning rate schedule, a warmup ratio of 0.1, and search weight decay over {0, 1e-6, 1e-2}. All models are trained in `bfloat16` with `FlashAttention-2` (Dao, 2023) enabled. For MIDUS-HML, to encourage balanced and sparse memory usage (Lample et al., 2019) while limiting additional hyperparameter tuning, we fix the learning rates of key and value parameters to the maximum learning rate without scheduling and set their weight decay to zero.

For Llama-3.2-1B on the `FineWeb-Edu` subset, we follow the hyperparameter configuration of prior work and apply the same search protocol to all baselines, including ours. During CPT, we search with a maximum learning rate of 1e-4, and during SFT, we fix the maximum learning rate to 1e-5. For Llama-3.1-8B on `FineWeb-Edu`, baseline results are taken from previous work, while for MIDUS-HML we fix the CPT learning rate to 5e-5 and search the SFT learning rate over {1e-6, 5e-6}. In the `MathPile` experiments, we use a learning rate of 1e-4 for all methods and both backbones. Unless otherwise stated (e.g., in ablation studies), we fix the global batch size to 64 and the sequence length to 2,048 tokens.

## B.2  DATASET DETAILS

Following Cao et al. (2025), we construct a 1.5B-token `FineWeb-Edu` subset by sampling from the CC-MAIN-2024-51 slice of `FineWeb-Edu` (Penedo et al., 2024) after the base model's pre-training cut-off. For SFT, we use the instruction-tuning datasets `Alpaca-GPT4` (52k examples) (Peng et al., 2023) and `Databricks-Dolly-15k` (15k examples)(Conover et al., 2023). To construct the `MathPile` subset, we extract math-related text from the six components of the original MATHPILE (Wang et al., 2024) corpus (Textbooks, Wikipedia, ProofWiki, CommonCrawl, StackExchange, and arXiv), excluding arXiv, resulting in approximately 1.1B tokens.

# C  IMPLEMENTATION DETAILS

While MIDUS shares the architectural skeleton with standard Transformer blocks, replacing dense FFNs with HML introduces distinct computational challenges. Unlike FFNs, which rely on computationally intensive but regular matrix multiplications (GEMMs), HML relies on sparse retrieval operations involving discrete addressing and scatter–gather patterns. Existing deep learning frameworks (e.g., PyTorch) provide highly optimized kernels for GEMM-based workloads but offer comparatively less optimization for this combination of sparse lookups, per-sample weighting, and index-based accumulation. As a result, naïve implementations can suffer from under-utilized memory bandwidth and severe atomic operation contention in CUDA kernels.

In this section, we describe three implementation-level optimizations employed in MIDUS to narrow this gap between theoretical and practical efficiency. These optimizations are orthogonal to the architectural design of HML and operate purely at the kernel and execution level. We note that further low-level engineering (e.g., fully fused CUDA kernels for the entire HML block) could yield additional gains beyond those reported in this work.

## C.1 FUSED CARTESIAN TOP-$k$ FOR EFFICIENT GENERATION

The standard PKM retrieval involves a hierarchical two-stage selection process to avoid evaluating scores for all $N = n^2$ slots. As described in Eq. 12 and Eq. 13, the model first identifies indices $I_{h,r}$ and $J_{h,r}$ by performing Top-$k$ on row and column sub-keys (each of length $n$), constructs $k^2$ candidate pairs $\Omega_{h,r} = I_{h,r} \times J_{h,r}$, and then performs a second Top-$k$ selection over the resulting pairwise scores. This row–column factorization reduces the theoretical complexity of PKM lookup from $\mathcal{O}(N)$ to $\mathcal{O}(n)$ per head by avoiding explicit scoring of all $n^2$ slots. However, in practice, the two-stage procedure incurs non-trivial kernel-launch overhead and irregular memory access patterns on GPUs, especially when the sequence length $s$ and batch size are small (e.g., during autoregressive generation). In this regime, the overhead of orchestrating multiple small kernels can dominate the arithmetic cost of the lookup itself.

To address this, we utilize a *Fused Cartesian Top-$k$* strategy for inference with short sequences. While Section 4.2 describes the conceptually standard two-stage PKM lookup, in the inference path we additionally provide an optimized variant that replaces the hierarchical selection with a single fused operation. Concretely, we compute the full sum of scores over the Cartesian grid $\mathcal{S}_{h,r} \in \mathbb{R}^{n \times n}$ such that

$$(\mathcal{S}_{h,r})_{ij} = S_{h,r}^{\text{row}}(i) + S_{h,r}^{\text{col}}(j), \tag{21}$$

and then perform a single Top-$k$ over the flattened $N$ scores:

$$\hat{\Omega}_{h,r} = \text{Top}_k\big(\text{flatten}(\mathcal{S}_{h,r})\big). \tag{22}$$

This computation is equivalent to scoring all $n^2$ product-key pairs, but it is executed as one large, highly parallel kernel that better utilizes the parallel processing capabilities of modern GPUs.

This fused strategy sacrifices the asymptotic $\mathcal{O}(n)$ lookup and returns to $\mathcal{O}(N)$ work. However, $n$ is fixed and modest in our setting, and during generation the sequence length and batch size are typically small. In this regime, the reduced kernel-launch overhead and improved hardware utilization yield lower end-to-end latency in practice. For longer sequences and larger batches, we revert to the standard two-stage PKM lookup described in Section 4.2.

## C.2 HEAD-WISE VALUE CACHING FOR INFERENCE

During training, HIVE dynamically generates head-specific values from a shared base value bank $\overline{V}$ via a linear projection $W_h$, as defined in Eq. 16:

$$M_{h,r}^{\text{HIVE}} = W_h \overline{M}_{h,r}, \tag{23}$$

where $\overline{M}_{h,r}$ denotes the aggregated base values retrieved from $\overline{V}$. This factorization substantially reduces the number of learnable parameters compared to maintaining an explicit, head-specific value table of size $N \times d_h$ for each head.

During inference, however, the model weights are frozen, and repeatedly applying $W_h$ to every retrieved value for each token and time step becomes redundant. We eliminate this overhead by pre-computing and caching the expanded value banks solely for the inference phase. Specifically, prior to decoding, we construct for each head $h$ a cached value bank

$$V_h^{\text{cache}} = \overline{V} W_h^\top \in \mathbb{R}^{N \times d_h}, \tag{24}$$

which is equivalent to applying $W_h$ to every row of $\overline{V}$. At run-time, the HML layer directly retrieves values from $V_h^{\text{cache}}$ using the selected memory indices, without recomputing the projection $W_h$.

This optimization trades an increase in memory usage for reduced computational latency. Importantly, it does not change the training-time parameterization or storage footprint of the model. The learned parameters remain factorized as $(\overline{V}, \{W_h\}_h)$, and the cached tables are constructed on demand only for inference. In addition, HML is inherently more memory-efficient than conventional memory layers in how it stores values. The shared bank $\overline{V} \in \mathbb{R}^{N \times d_h}$ lives in the per-head dimension $d_h = d_{\text{model}}/H$, and HML concatenates head-wise outputs only after retrieval. By contrast, linear or PKM-style memory layers typically store values directly in the full hidden dimension $d_{\text{model}}$. As a result, even after materializing the head-specific caches $\{V_h^{\text{cache}}\}_h$, the total value footprint remains significantly smaller than that of these baselines and negligible relative to the 1B/8B backbone. All

inference-side efficiency metrics reported in our experiments (e.g., peak GPU memory during decoding, generation throughput, and prefill time) are measured with value caching enabled, and the peak-memory numbers still show that HML is the most memory-efficient depth up-scaling design among the baselines.

## C.3 GRADIENT CONFLICT MITIGATION VIA DEDUPLICATION

A critical bottleneck in training memory-augmented networks arises during the backward pass of the embedding retrieval (i.e., value aggregation). When multiple tokens in a batch access the same memory slot index, standard implementations (e.g., `torch.nn.EmbeddingBag` or `scatter_add`) suffer from *atomic operation contention*. The GPU hardware serializes concurrent `atomicAdd` requests to the same memory address, causing severe performance degradation.

In addition, for the specific combination of `bfloat16` values and `per_sample_weights` used in our memory layers, current PyTorch kernels do not provide an efficient backward implementation that meets our requirements. To address both issues, we implement a custom autograd function that employs a "Deduplication and Pre-aggregation" strategy. This backward scheme applies to any memory layer that aggregates values from an indexable table (e.g., PKM, linear memory, and HML).

As outlined in Algorithm 1, our method first expands gradients to per-token form, then deduplicates indices within the batch, and finally performs local aggregation before a single global update. This consolidates all contributions to a given memory slot into a single write, greatly reducing atomic contention and improving effective memory-bandwidth utilization.

---

**Algorithm 1** Deduplication and Pre-aggregation Backward Pass

---

**Require:** Gradient w.r.t. output $G_{\text{out}} \in \mathbb{R}^{B \times D}$, indices $Idx \in \mathbb{Z}^{B \times K}$, weights $W \in \mathbb{R}^{B \times K}$,
   value-table size $V_{\text{size}}$, $B$: batch size, $K$: # retrieved slots per token, $D$: value dimension
 1: **Step 1: Broadcast and Expand Gradients**
 2: $G_{\text{bcast}} \leftarrow G_{\text{out}}.\text{unsqueeze}(1).\text{expand}(B, K, D)$
 3: $G_{\text{token}} \leftarrow \left(G_{\text{bcast}} \odot W.\text{unsqueeze}(-1)\right).\text{flatten}(0, 1) \ \{G_{\text{token}} \in \mathbb{R}^{BK \times D}\}$
 4: $Idx_{\text{flat}} \leftarrow Idx.\text{flatten}() \ \{Idx_{\text{flat}} \in \mathbb{Z}^{BK}\}$
 5: **Step 2: Deduplication (Conflict Resolution)**
 6: $Idx_{\text{uniq}}, Idx_{\text{inv}} \leftarrow \text{torch.unique}(Idx_{\text{flat}}, \text{return\_inverse=True})$
 7: **Step 3: Pre-aggregation (Local Reduction)**
 8: {Accumulate gradients locally for each unique index}
 9: $U \leftarrow Idx_{\text{uniq}}.\text{size}(0)$
10: $G_{\text{agg}} \leftarrow \text{torch.zeros}(U, D, \text{dtype=bfloat16})$
11: $G_{\text{agg}}.\text{index\_add\_}(0, Idx_{\text{inv}}, G_{\text{token}})$
12: **Step 4: Global Update**
13: {Single aggregated update per memory slot}
14: $G_{\text{embed}} \leftarrow \text{torch.zeros}(V_{\text{size}}, D, \text{dtype=bfloat16})$
15: $G_{\text{embed}}.\text{index\_add\_}(0, Idx_{\text{uniq}}, G_{\text{agg}})$
16: **return** $G_{\text{embed}}$

---

In practice, this backward scheme reproduces the gradients of the naïve implementation but avoids repeated atomic updates to the same memory location. The resulting kernel significantly accelerates the training of memory layers. (Gradients with respect to the per-sample weights $W$ follow the standard `EmbeddingBag` rule and are omitted here for brevity.)

## D  DUS PLACEMENT POLICY

DUS placement policy is a key design choice that affects not only CPT performance but also training efficiency. The Llama-3.2-1B and Llama-3.1-8B base models consist of 16 and 32 Transformer layers, respectively. For all DUS and MIDUS variants, we up-scale these backbones with 8 and 16 additional blocks, yielding total depths of 24 and 48 layers. The indices of the expanded blocks under each DUS placement policy are as follows.

**Up-scaled Llama-3.2-1B (24 layers).**

- *Top-heavy (OpT-DeUS)*: {8, 10, 12, 14, 16, 18, 20, 22}
- *Llama Pro*: {2, 5, 8, 11, 14, 17, 20, 23}
- *Distributed*: {1, 4, 7, 10, 13, 16, 19, 22}
- *Bottom-heavy*: {0, 2, 4, 6, 8, 10, 12, 14}

**Up-scaled Llama-3.1-8B (48 layers).**

- *Top-heavy (OpT-DeUS)*: {16, 18, 20, 22, 24, 26, 28, 30, 32, 34, 36, 38, 40, 42, 44, 46}
- *Llama Pro*: {2, 5, 8, 11, 14, 17, 20, 23, 26, 29, 32, 35, 38, 41, 44, 47}
- *Distributed*: {1, 4, 7, 10, 13, 16, 19, 22, 25, 28, 31, 34, 37, 40, 43, 46}
- *Bottom-heavy*: {0, 2, 4, 6, 8, 10, 12, 14, 16, 18, 20, 22, 24, 26, 28, 30}

If we focus only on the choice of DUS placement policy, *Top-heavy* is the most training-efficient design, since its top-heavy placement reduces both the amount of backward computation and the activation memory required for expanded blocks. *Llama Pro* is also more efficient than *distributed* placement for the same reason. In contrast, MIDUS-HML achieves its best performance under the *Distributed* policy, yet it still delivers substantially better overall efficiency than the DUS baselines. Moreover, MIDUS-HML can adopt the *Top-heavy* policy to push efficiency further, even in this extreme setting, its performance remains comparable to, or better than, that of the DUS baselines. See Table 11 and Appendix G for further ablation studies.

# E  SFT RESULTS ON DATABRICKS-DOLLY-15K

Table 6: SFT results on the Databricks-Dolly-15k dataset (Conover et al., 2023) with Llama-3.2-1B across DUS baselines and MIDUS.

| | Methods | Perplexity ↓ | | Zero-shot Accuracy ↑ | | | | | | |
| | | Wiki-PPL | ARC | LogiQA | Wino | CSQA | BoolQ | PIQA | MMLU | Average |
|---|---|---|---|---|---|---|---|---|---|---|
| CPT-1B | Base | 13.07 | 68.64 | 21.35 | 60.30 | 27.52 | 63.27 | **75.24** | 30.49 | 49.55 |
| | SOLAR | 13.19 | **69.40** | 23.04 | 59.04 | 27.52 | 60.03 | 75.14 | 30.77 | 49.28 |
| | Llama Pro | 12.47 | 68.10 | **23.81** | 60.54 | 41.93 | 63.70 | **75.24** | 34.03 | 52.48 |
| | LESA | 11.77 | 66.46 | 23.66 | 60.77 | 49.06 | 64.98 | 74.86 | 38.22 | 54.00 |
| | OpT-DeUS | 11.78 | 67.21 | **23.81** | 61.48 | **49.14** | 63.76 | 75.19 | 37.83 | 54.06 |
| | Avg-DeUS | 12.10 | 67.51 | 22.73 | 60.30 | 46.60 | 65.02 | 74.37 | 36.79 | 53.33 |
| | MIDUS-HML | **11.72** | 66.62 | 22.12 | **61.72** | 48.65 | **66.51** | 75.14 | **38.59** | **54.19** |

# F  ABLATION STUDIES

## F.1  ABLATION STUDY ON THE NUMBER OF MEMORY

Table 7: Effect of total memory slots on performance. The 66K, 262K, and 1M settings correspond to $n = 16, 32, 64$ in the PKM factorization (so $N = n^2$).

| | Perplexity ↓ | | Zero-shot Accuracy ↑ | | | | | | |
| # Memories | Wiki-PPL | ARC | LogiQA | Wino | CSQA | BoolQ | PIQA | MMLU | Average |
|---|---|---|---|---|---|---|---|---|---|
| 66K ($n = 16$) | **11.63** | 66.12 | 22.89 | 60.06 | 46.27 | 65.02 | 74.81 | 36.74 | 53.13 |
| 262K ($n = 32$) | 11.64 | 65.95 | **23.35** | 60.85 | **47.17** | 65.44 | 74.76 | 36.41 | 53.42 |
| 1M ($n = 64$) | 11.64 | **66.16** | 23.20 | **61.56** | 46.27 | 65.29 | 75.08 | **36.91** | **53.50** |

In Table 7, we vary the total number of product-key memories from tens of thousands to one million, i.e., $n=16, 32, 64$ in the PKM factorization. Average zero-shot accuracy rises steadily with capacity, while perplexity stays essentially flat. We hypothesize that larger tables mainly help tasks that benefit from retrieving facts or patterns, without disrupting the core language-modeling behavior. MIDUS gains an additional axis to scale quality, and rather than adding more Transformer blocks, we can improve results by increasing the number of memories.

## F.2 Ablation study on Top-k

Table 8: Effect of the number of Top-$k$ for Llama-3.2-1B with MIDUS-HML.

| | Perplexity ↓ | Zero-shot Accuracy ↑ | | | | | | | |
|---|---|---|---|---|---|---|---|---|---|
| Top-$k$ | Wiki-PPL | ARC | LogiQA | Wino | CSQA | BoolQ | PIQA | MMLU | Average |
| 1 | 11.67 | 65.74 | 21.97 | 60.54 | 46.19 | 64.95 | 74.86 | 36.95 | 53.03 |
| 2 | **11.63** | 65.87 | **23.20** | 60.69 | 46.11 | 65.11 | 74.54 | 37.04 | 53.22 |
| 4 | 11.64 | **66.16** | **23.20** | 61.56 | 46.27 | **65.29** | 75.08 | 36.91 | **53.50** |
| 8 | 11.64 | 66.04 | 23.04 | 60.46 | **46.93** | 64.98 | 75.03 | **37.11** | 53.37 |

In Table 8, we vary the number of retrieved memory slots per token from $k=1$ to $k=8$ while keeping the memory size and all other hyperparameters fixed. Average zero-shot accuracy improves steadily as $k$ increases from 1 to 4, with modest or no further gains at $k=8$, and perplexity remains essentially unchanged across all settings. We therefore use $k=4$ as the default Top-$k$ setting in all other experiments.

## F.3 Ablation study on the learning rate and weight decay

Table 9: Effect of learning rate (LR) and weight decay (WD) on the performance of Llama-3.2-1B with MIDUS-HML

| | | Perplexity ↓ | Zero-shot Accuracy ↑ | | | | | | | |
|---|---|---|---|---|---|---|---|---|---|---|
| LR | WD | Wiki-PPL | ARC | LogiQA | Wino | CSQA | BoolQ | PIQA | MMLU | Average |
| 1e-4 | 1e-6 | 11.64 | 66.16 | **23.20** | **61.56** | 46.27 | 65.29 | **75.08** | 36.91 | **53.50** |
| 1e-4 | 0 | 11.64 | 66.25 | 21.97 | 60.77 | 46.60 | 65.32 | 75.03 | 36.99 | 53.27 |
| 1e-4 | 1e-2 | **11.62** | 65.78 | 23.04 | 59.91 | 45.45 | **65.35** | 74.81 | **37.25** | 53.09 |
| 1e-5 | 1e-6 | 11.64 | 65.57 | 21.51 | 61.01 | 46.85 | 63.58 | 74.48 | 37.12 | 52.87 |
| 5e-5 | 1e-6 | 11.63 | **66.46** | 22.89 | 60.14 | **47.58** | 64.86 | 74.54 | 37.23 | 53.39 |

In Table 9, we vary the learning rate and weight decay over a range of reasonable values while keeping all other settings fixed. Across these configurations, perplexity remains tightly clustered around 11.62–11.64 and the average zero-shot accuracy stays in a narrow band, indicating that MIDUS-HML is relatively robust to these optimization hyperparameters. For the memory layer itself, we use the same learning rate but apply it as a fixed rate without a scheduler, and we set the weight decay for the key–value parameters to zero.

## F.4 Ablation study on Structure of HML Memory Layer

Table 10: Ablation studies on the structure of HML.

| | Perplexity ↓ | Zero-shot Accuracy ↑ | | | | | | | |
|---|---|---|---|---|---|---|---|---|---|
| Method | Wiki-PPL | ARC | LogiQA | Wino | CSQA | BoolQ | PIQA | MMLU | Average |
| MIDUS-HML | **11.64** | 66.16 | 23.20 | **61.56** | 46.27 | **65.29** | 75.08 | 36.91 | 53.50 |
| w/ BatchNorm1D | 11.74 | **66.54** | 22.43 | 60.77 | **47.34** | 64.80 | 74.92 | 36.94 | 53.39 |
| w/ LayerNorm | 11.68 | 66.08 | 22.43 | **61.56** | 46.68 | 65.17 | 75.19 | **37.34** | 53.49 |
| w/ Residual | **11.64** | 66.04 | 22.89 | 61.48 | 46.36 | 65.17 | **75.35** | 36.75 | 53.43 |
| w/ Output Projection | **11.64** | 66.12 | **23.66** | 61.09 | 46.52 | 65.23 | 74.86 | 37.07 | **53.51** |

In Table 10, we evaluate four variants of HML: adding BatchNorm1d to the queries (as in PKM), applying a LayerNorm normalization to the queries, inserting an internal residual path $a' = x + \text{Attn}'(\text{LN}_{\text{Attn}}(x))$ that passes $\text{Attn}'$, and enabling an attention layer with output projection. We hypothesize that query normalization may blur the relative score magnitudes that drive top-$k$ selection, and that the internal residual may distort the head-specific signals that serve as queries, thereby weakening the retrieved signal. Adding an output projection yields only a marginal gain and does not change the overall picture. Using head embeddings directly as queries and retaining a pure memory residual $M(x) = x + m$ offers the most favorable balance, achieving the lowest perplexity and the highest average accuracy in Table 10.

## F.5 ABLATION STUDY ON DUS PLACEMENT POLICY.

Table 11: MIDUS-HML under different DUS policies. GPU Memory and Time are training requirements. *Top-heavy* places expanded blocks toward the top, *Distributed* interleaves, and *Bottom-heavy* places them near the input.

| DUS Policy | GB ↓ GPU Memory | s/iter ↓ Time | Perplexity ↓ Wiki-PPL | Zero-shot Accuracy ↑ ARC | LogiQA | Wino | CSQA | BoolQ | PIQA | MMLU | Average |
|---|---|---|---|---|---|---|---|---|---|---|---|
| *Top-heavy* | **24.1** | **3.89** | 11.63 | 66.12 | 22.27 | 61.25 | **46.44** | 64.28 | 74.92 | 36.23 | 53.07 |
| *Distributed* | 28.0 | 4.56 | 11.64 | **66.16** | **23.20** | **61.56** | 46.27 | **65.29** | 75.08 | **36.91** | **53.50** |
| *Bottom-heavy* | 28.6 | 4.67 | 11.60 | 66.12 | 21.81 | 60.30 | 45.05 | 65.05 | **75.30** | 36.35 | 52.85 |

Table 11 shows the efficiency and performance of MIDUS-HML with Llama-3.2-1B under different DUS policies. *Top-heavy* placement, as used by OpT-DeUS (Cao et al., 2025),is most efficient in memory and time, since many frozen lower blocks can be skipped during backpropagation. *Bottom-heavy* placement slightly improves perplexity but does not maximize average zero-shot accuracy. The *distributed* policy yields the best overall accuracy by injecting head-structured retrieval at multiple depths.

## G ADDITIONAL EFFICIENCY EXPERIMENTS

We measure all efficiency metrics on a single NVIDIA RTX 6000 Blackwell GPU. For training-related metrics, Table 4 and Figure 3 use a global batch size of 16 with 16 gradient-accumulation steps, while Table 12 and Figure 5 report results with a global batch size of 64 and 16 gradient-accumulation steps. Unless otherwise noted, the sequence length is fixed to 2048 in these efficiency experiments. For inference-related metrics, we report peak GPU memory and generation throughput measured during the generation stage.

Figure 6 further examines efficiency as a function of the global batch size. We fix the gradient-accumulation steps to 16 and the sequence length to 2048, and vary only the global batch size. For Llama-3.1-8B, we omit the training-time entries for Llama Pro and OpT-DeUS at a global batch size of 64, since they exceed the available GPU memory on a single RTX 6000 Blackwell GPU. Across Figure 6 and 7, which cover a range of batch sizes and sequence lengths, MIDUS-HML (*Dist.*) consistently achieves training times that are faster than or comparable to those of the DUS baselines. Notably, MIDUS-HML (*Top.*), which adopts the same DUS placement policy as OpT-DeUS, is substantially faster than OpT-DeUS. In all cases, MIDUS-HML also requires less peak GPU memory than any of the DUS baselines. The Llama-3.1-8B results in particular indicate that MIDUS-HML becomes even more effective on larger and deeper backbones.

Conversely, on smaller backbones, DUS can still be faster for generation or for short sequence lengths. Table 11 and Figure 7, which report efficiency results on Llama-3.2-1B, show that DUS achieves lower latency than MIDUS-HML in this regime. However, MIDUS-HML still uses fewer parameters and requires less peak GPU memory than the DUS baselines, and it regains a clear advantage in training and generation speed as the sequence length increases. This crossover behavior between DUS and MIDUS-HML is largely driven by current implementation overheads rather than a fundamental limitation. As discussed in Appendix C, MIDUS-HML still relies on non–fully optimized low-level kernels for its memory layers, and we therefore expect the crossover point to move further in favor of MIDUS-HML as these kernels are more aggressively optimized.

Table 12: Efficiency of DUS and MIDUS for Llama-3.2-1B, with the number of parameters (trainable/total), GPU memory (train/inference), training time, and generation throughput.

| | Method | B ↓ Params | GB ↓ GPU Memory | s/iter ↓ Train | Tokens/s ↑ Throughput |
|---|---|---|---|---|---|
| | Base | 1.23 / 1.23 | 43.9 / 10.4 | 4.74 | 144.4 |
| DUS | Llama Pro | 0.49 / 1.72 | 42.8 / 14.5 | 5.26 | **99.1** |
| | SOLAR | 1.72 / 1.72 | 49.0 / 14.5 | 6.48 | **99.1** |
| | OpT-DeUS | 0.49 / 1.72 | 38.3 / 14.5 | 4.58 | **99.1** |
| MIDUS | Linear | 0.35 / 1.59 | 46.4 / 14.2 | 10.79 | 79.5 |
| | PKM | 0.19 / 1.42 | 43.2 / 14.0 | 8.42 | 91.5 |
| | HML | **0.05 / 1.29** | **30.1 / 13.5** | **4.56** | 98.6 |

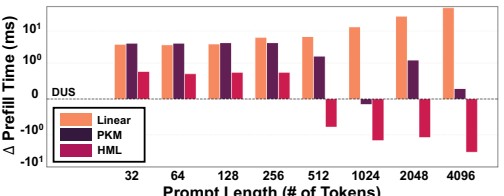

Figure 5: Change in prefill time versus prompt length for each memory layer relative to DUS, where negative Δ Prefill Time indicates faster prefill time than DUS.

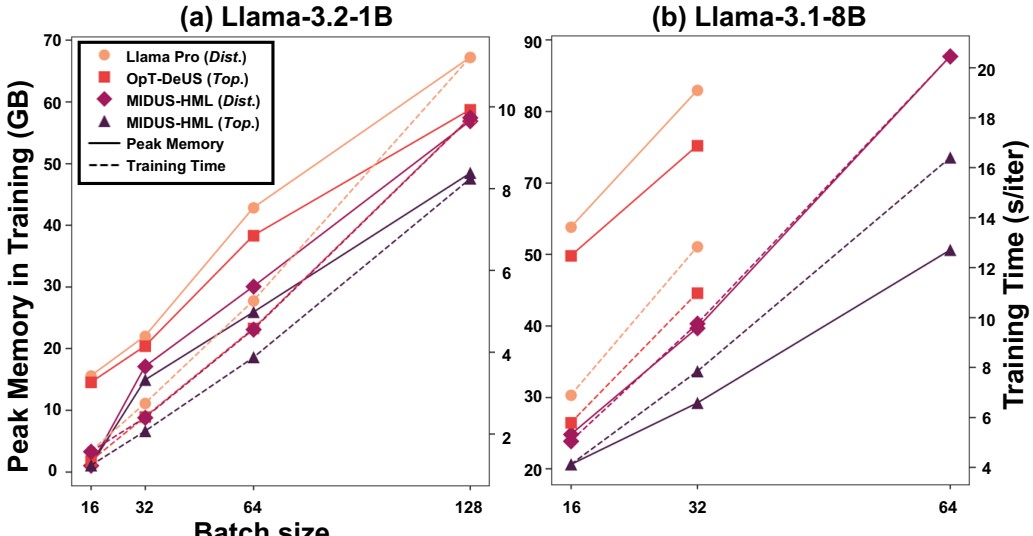

Figure 6: Efficiency of DUS baselines and MIDUS-HML as a function of batch size. Here, *Dist.* denotes the *Distributed* and *Top.* the *Top-heavy* DUS placement policy.

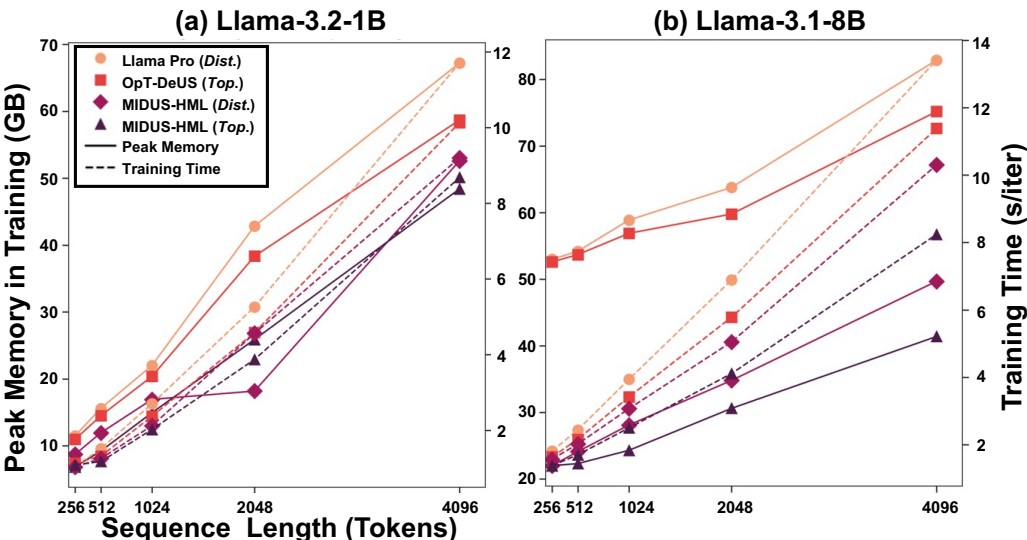

Figure 7: Efficiency of DUS baselines and MIDUS-HML as a function of sequence length. Here, *Dist.* denotes the *Distributed* and *Top.* the *Top-heavy* DUS placement policy.

