# OpenReview forum: "MIDUS: Memory-Infused Depth Up-Scaling"
_ICLR.cc/2026/Conference — Submitted to ICLR 2026_

### Official Review · Reviewer_NZXf · 2025-10-23

**Soundness:** 3
**Presentation:** 3
**Contribution:** 3
**Rating:** 6
**Confidence:** 2

**Summary:**

This paper introduces MIDUS (Memory-Infused Depth Up-Scaling), a method for Depth Up-Scaling (DUS) that replaces FFN layers with a "Memory Block". The main contributions include the Head-wise Memory Layer (HML), which assigns memory per attention head, and efficient storage mechanisms (PKM for keys, HIVE for values). The authors report that this method achieves performance comparable to or better than DUS baselines while adding significantly fewer trainable parameters and offering comparable or faster computation times.

**Strengths:**

1. MIDUS can achieve strong performance for DUS using siginifantly fewer parameters
2. Ingenious design of the memory block. Using two separate $K$ to reduce computational cost and parameter count.

**Weaknesses:**

1. Limited scope. The paper only explore the interleaving memory blocks. And other DUS policies are not discussed which may also enhance thier efficency.
2. New hyperparameter complexity. MIDUS introduces new and non-trivial design choices such as the memory size, which increases the difficulty of finding the optimal hyperparameters.

**Questions:**

1. The paper explicitly avoids stacking Memory Blocks. Was this simply out of scope, or did you find any evidence that stacking $M^{HML}$ blocks leads to instability?
2. The decomposition of $K_h$ into two parts for the two-dimensional search is an effective strategy. Has this approach been extended to higher dimensions, such as a three-dimensional search?

typo: line 81-82, "\\$H\\$"

---

> ### Author Response · Authors · 2025-11-29
>
> Thank you for the thoughtful and constructive review. In the revision we (i) make our use of DUS placement policies more explicit and compare MIDUS–HML under Top-heavy / Distributed / Bottom-heavy allocations, (ii) clarify the additional hyperparameters introduced by the memory layer and provide sensitivity analyses for memory size and top-k retrieval, and (iii) discuss stacking memory blocks and the possible extension of our two-dimensional product-key design to higher-dimensional searches. Below we respond to each of your points in detail.
>
> ---
> > **W1) Limited scope. The paper only explores interleaving memory blocks. And other DUS policies are not discussed which may also enhance their efficiency.**
>
> We agree that the choice of DUS policy is important for understanding how MIDUS–HML behaves in practice. In the main text, we focused on the interleaving setup because it is the standard configuration used by existing DUS baselines, and this makes it easier to isolate the effect of replacing FFNs with head-wise memory.
>
> In the revision, we now discuss alternative DUS placement policies more explicitly.
> - **Appendix F.5-Table 11) Ablation study on the DUS placement policy**
>
> | DUS Policy     | GPU Memory (GB ↓) | Time (s/iter ↓) | Wiki-PPL ↓ |     ARC ↑ |  LogiQA ↑ |    Wino ↑ |    CSQA ↑ |   BoolQ ↑ |    PIQA ↑ |    MMLU ↑ | Average ↑ |
> | -------------- | ----------------: | --------------: | ---------: | --------: | --------: | --------: | --------: | --------: | --------: | --------: | --------: |
> | *Top-heavy*    |          **24.1** |        **3.89** |    *11.63* |   *66.12* |   *22.27* |   *61.25* | **46.44** |     64.28 |     74.92 |     36.23 |   *53.07* |
> | *Distributed*  |            *28.0* |          *4.56* |      11.64 | **66.16** | **23.20** | **61.56** |   *46.27* | **65.29** |   *75.08* | **36.91** | **53.50** |
> | *Bottom-heavy* |              28.6 |            4.67 |    *11.60* | **66.12** |     21.81 |     60.30 |     45.05 |   *65.05* | **75.30** |   *36.35* |     52.85 |
>
> - Section 5.1 and Appendix D describe and compare three DUS policies for MIDUS–HML, **Top-heavy**, **Distributed**, and **Bottom-heavy**, and explain how they differ in terms of where memory blocks are inserted and how this affects training-time efficiency.
> - Appendix F.5 (Table 11) reorganizes the results for these policies on LLaMA-3.2-1B, summarizing their perplexity, zero-shot accuracy, training time, and GPU memory side by side.
>
> Across all three policies, MIDUS–HML remains comparable to or better than DUS baselines in terms of accuracy, while using fewer trainable parameters and lower training memory. Importantly, our main configuration for MIDUS–HML uses the *Distributed* policy, which is less favorable for efficiency than the *Top-heavy* schedule adopted by OpT-DeUS,since memory is spread across multiple depths rather than concentrated at the top. Even under this less efficiency-oriented policy, Table 4 and Table 12, together with Figures 3,5,6 and 7, show that MIDUS–HML still achieves lower training and inference memory and comparable or better latency than OpT-DeUS at both 1B and 8B scales. Moreover, if we adopt the same *Top-heavy* DUS policy as OpT-DeUS, MIDUS–HML attains clearly higher efficiency while maintaining comparable performance. Overall, these results indicate that the gains of MIDUS–HML are not tied to a single DUS policy, and that the method remains robust across reasonable choices of depth placement.

---

> > ### Author Response · Authors · 2025-11-29
> >
> > > **W2) New hyperparameter complexity. MIDUS introduces new and non-trivial design choices such as the memory size, which increases the difficulty of finding the optimal hyperparameters.**
> >
> > We agree that additional hyperparameters would be problematic if MIDUS–HML were highly sensitive to them. To address this, Appendix F systematically varies the main memory-related choices and optimization settings, and we observe that performance changes smoothly rather than exhibiting sharp or brittle behavior.
> >
> > - **Memory size (Appendix F.1).**
> > We vary the total number of memory slots per head across three scales and observe that larger capacity consistently yields higher average zero-shot accuracy, while perplexity stays essentially unchanged. This indicates that increasing memory size is beneficial but not critical to tune precisely—MIDUS–HML performs well across a reasonable range of memory sizes. We fix the memory size as $n=64$ for all other experiments.
> > - **Appendix F.1-Table 7) Sensitivity to the memory size n**
> >
> > |  # Memories |  Wiki-PPL |    ARC    |   LogiQA  |    Wino   |    CSQA   |   BoolQ   |   PIQA  |    MMLU   |  Average  |
> > | ----------: | :-------: | :-------: | :-------: | :-------: | :-------: | :-------: | :-----: | :-------: | :-------: |
> > |  66K (n=16) | **11.63** |  *66.12*  |   22.89   |   60.06   |  *46.27*  |   65.02   |  74.81  |  *36.74*  |   53.13   |
> > | 262K (n=32) |  *11.64*  |   65.95   | **23.35** |  *60.85*  | **47.17** | **65.44** |  74.76  |   36.41   |  *53.42*  |
> > |   1M (n=64) |  *11.64*  | **66.16** |  *23.20*  | **61.56** |  *46.27*  |  *65.29*  | *75.08* | **36.91** | **53.50** |
> >
> > - **Top-k retrieval (Appendix F.2).**
> > We vary the number of retrieved slots per token Top-$k$ and find that performance is stable across a reasonable range. Accuracy improves when moving from very small to moderate values of $k$, while perplexity changes only slightly. We therefore fix a single moderate value of $k=4$ in all main experiments, without per-task tuning.
> >
> > - **Appendix F.2-Table 8) Sensitivity to top-k**
> >
> > | top-k |  Wiki-PPL |    ARC    |   LogiQA  |    Wino   |    CSQA   |   BoolQ   |   PIQA  |    MMLU   |  Average  |
> > | ----: | :-------: | :-------: | :-------: | :-------: | :-------: | :-------: | :-----: | :-------: | :-------: |
> > |     1 |   11.67   |   65.74   |   21.97   |   60.54   |   46.19   |   64.95   |  74.86  |   36.95   |   53.03   |
> > |     2 | **11.63** |   65.87   | **23.20** |  *60.69*  |   46.11   |  *65.11*  |  74.54  |  *37.04*  |   53.22   |
> > |     4 |  *11.64*  | **66.16** | **23.20** | **61.56** |  *46.27*  | **65.29** | *75.08* |   36.91   | **53.50** |
> > |     8 |  *11.64*  |  *66.04*  |  *23.04*  |   60.46   | **46.93** |   64.98   | *75.03* | **37.11** |  *53.37*  |
> >
> >
> > - **Learning rate and weight decay (Appendix F.3).**
> >   We explore several combinations of learning rate and weight decay, including variants where the memory-layer parameters use different regularization. Across these settings, both perplexity and average zero-shot accuracy stay within a narrow band, indicating that MIDUS–HML is not overly sensitive to these optimization choices. For the memory keys/values, we use a fixed learning rate without weight decay.
> > - **Appendix F.3-Table 9) Ablation studies on learning rate and weight decay**
> >
> > | LR   | WD    | Wiki-PPL |   ARC    | LogiQA  |  Wino   |  CSQA  | BoolQ  |  PIQA  | MMLU  | Average |
> > |------|-------|----------|----------|---------|---------|--------|--------|--------|-------|---------|
> > | 1e-4 | 1e-6  | 11.64    | 66.16    | **23.20** | **61.56** | 46.27  | 65.29  | **75.08** | 36.91 | **53.50** |
> > | 1e-4 | 0     | 11.64    | *66.25*  | 21.97   | 60.77   | 46.60  | *65.32* | *75.03* | 36.99 | 53.27  |
> > | 1e-4 | 1e-2  | **11.62** | 65.78   | *23.04* | 59.91   | 45.45  | **65.35** | 74.81 | **37.25** | 53.09  |
> > | 1e-5 | 1e-6  | 11.64    | 65.57    | 21.51   | *61.01* | *46.85* | 63.58  | 74.48  | 37.12 | 52.87  |
> > | 5e-5 | 1e-6  | *11.63*  | **66.46** | 22.89  | 60.14   | **47.58** | 64.86 | 74.54  | *37.23* | *53.39* |
> >
> >
> > These ablations show that the additional hyperparameters introduced by MIDUS–HML behave in a smooth and interpretable way.

---

> > > ### Author Response · Authors · 2025-11-29
> > >
> > > > **Q1) The paper explicitly avoids stacking Memory Blocks. Was this simply out of scope, or did you find any evidence that stacking blocks leads to instability?**
> > >
> > > In MIDUS–HML, the memory blocks are not designed to form a standalone stacked tower. Architecturally, each HML block is *interleaved* between two Transformer blocks and is always applied to the output of a preceding Transformer layer before passing the result to the next one. The goal of the memory layer in this design is to *augment* the contextual representation produced by the previous Transformer block with head-wise retrieved information, so that the subsequent Transformer block can exploit these additional signals.
> > >
> > > Concretely, given the output $x_\ell$ of Transformer layer $\ell$, the HML block computes a head-wise retrieval $m_\ell(x_\ell)$ and returns $x_{\ell+1}^{\text{(mem)}} = x_\ell + m_\ell(x_\ell)$, which is then fed into Transformer layer $\ell+1$. The memory block is therefore conceptually “in-between” Transformer layers rather than a unit we intend to stack repeatedly in isolation.
> > >
> > > For this reason, we did not pursue architectures where multiple Memory blocks are placed consecutively without an intervening Transformer layer, not because we observed intrinsic instability, but because such stacking is not aligned with the intended role of HML as an interleaved-augmentation operator of Transformer representations. Exploring deeper stacks of memory-only blocks, e.g., memory towers on top of the backbone, is therefore orthogonal to our main design and remains an interesting direction for future work rather than a limitation due to instability.
> > >
> > > ---
> > >
> > > > **Q2) The decomposition of keys into two parts for the two-dimensional search is an effective strategy. Has this approach been extended to higher dimensions, such as a three-dimensional search?**
> > >
> > > Our key design follows the standard product-key memory formulation. Each key is factorized into two sub-keys, and retrieval is performed via a two-dimensional product-key search. This 2D factorization can be understood as an **approximate realization** of the full Top-$k$ search over unfactored keys [1]. Instead of computing similarities in the full key space, we perform two lower-dimensional searches over the sub-key spaces and then combine their candidates.
> > >
> > > In principle, one could extend this idea to a 3D (or higher-dimensional) product-key formulation, where each key is split into three sub-keys and retrieval operates over a 3D Cartesian product. This would further increase the addressable capacity for a fixed sub-key size and could lead to additional efficiency gains in the lookup phase.
> > >
> > > We did not adopt a 3D product-key variant in this work for two main reasons. First, product-key search is already an approximation to the ideal linear scan over all keys, and moving from 2D to 3D deepens this factorization and further constrains how the effective Top-$k$ set is constructed from sub-key scores. While this could improve efficiency, it raises a methodological question of whether a 3D factorization still approximates the intended head-wise Top-$k$ behavior faithfully in practice, properly addressing this would require a dedicated study of retrieval quality and gradient flow under higher-dimensional factorization, which lies beyond our current scope.
> > >
> > > Second, the focus of this paper is not to optimize PKM itself, but rather to study memory-infused depth up-scaling and, in particular, the effect of assigning independent memory banks per attention head through HML. Our goal is to understand how head-wise memory changes representation usage, head importance, and efficiency relative to DUS baselines. For this purpose, the standard 2D product-key design already provides sufficient capacity and efficiency [2], and our ablations suggest that performance is not bottlenecked by key dimensionality. We therefore view higher-dimensional, e.g., 3D, product-key search as a natural and interesting extension for future work aimed at further improving the efficiency–accuracy trade-off of memory layers, while in this paper we deliberately keep the key search mechanism at the well-understood 2D level to more cleanly isolate the contribution of the MIDUS–HML architecture.
> > >
> > > ---
> > > [1] Huang, Zihao, et al. "Ultra-Sparse Memory Network." The Thirteenth International Conference on Learning Representations.
> > > [2]  Lample, Guillaume, et al. "Large memory layers with product keys." Advances in Neural Information Processing Systems 32 (2019).

---

> > > > ### Author Response · Authors · 2025-12-01
> > > > **Remind of Revisions Addressing Reviewer Concerns**
> > > >
> > > > We appreciate this constructive review. In the revision, we make the DUS placement policies explicit (comparing Top-heavy, Distributed, and Bottom-heavy), summarize ablations on memory-related hyperparameters to show that MIDUS–HML is robust to these choices, and clarify both the intended interleaved role of memory blocks and the possible extension to higher-dimensional product-key search. We remain fully available for any further questions or discussion about these revisions, if the review process allows.

---

### Official Review · Reviewer_9rn7 · 2025-10-31

**Soundness:** 3
**Presentation:** 2
**Contribution:** 2
**Rating:** 2
**Confidence:** 4

**Summary:**

The paper proposes MIDUS (Memory-Infused Depth Up-Scaling), a drop-in alternative to standard depth up-scaling (DUS) that replaces duplicated FFN layers with Memory blocks. Each Memory block combines an attention layer without the output projection (Attn′) and a Head-wise Memory Layer (HML) that performs per-head product-key retrieval with an efficient value factorization (HIVE). The design preserves identity at initialization by zero-initializing memory values and routing the retrieved signal through a residual path, so the expanded model initially matches the base model’s outputs. Experiments on Llama-3.2-1B with 8 inserted blocks demonstrate lower perplexity and improved average zero-shot accuracy compared to strong DUS baselines, while reducing trainable parameters and training-time memory. Figure 1 contrasts DUS vs. MIDUS; Figure 2 (p.5) details the six-step Memory-block pipeline.

**Strengths:**

1. Clear, modular design (Memory block + HML + HIVE) that can be inserted wherever DUS would add FFNs; identity-preserving init is well motivated.

2. Consistent gains across CPT and SFT with lower parameter/memory cost than DUS, plus ablations and placement analysis.

3. Clarity & completeness: math formalization, stepwise diagram (p.5), and a reproducibility-friendly recipe; an anonymized code link is included.

**Weaknesses:**

0. Some claims, especially those related to the major motivation, that existing methods rely "on dense feed-forward networks", are not accurate. For example, papers [1][2] are using "mixture of depth" like a sparse module for up-scaling. There is no discussion on the difference between these works, and they were not included as baselines.

1. All results use a 1B backbone. Claims about general-purpose LLM scaling would be stronger with a 7B-class (or larger) model and at least one instruction-tuned setting beyond Alpaca-GPT-4.

2. Iteration times are reported, but sensitivity to sequence length (e.g., 8k–32k) isn’t analyzed; PKM’s two-stage top-k might have different break-even points. (Tables 2–4 give per-iter stats only)

3. Helpful ablations are included, but further disentangling the impact of (i) Attn′ vs. full MHA, (ii) exact k and n choices, and (iii) HIVE’s parameterization per head would clarify where the gains come from.

4. The benchmark suite is knowledge-centric; evaluating on reasoning-heavy or long-context tasks would test whether HML helps beyond factual retrieval

[1] Raposo, David, et al. "Mixture-of-depths: Dynamically allocating compute in transformer-based language models." arXiv preprint arXiv:2404.02258 (2024).

[2] Tan Z, Dong D, Zhao X, et al. Dlo: Dynamic layer operation for efficient vertical scaling of llms[J]. arXiv preprint arXiv:2407.11030, 2024.

**Questions:**

1. Can you detail the difference/limitation of the related works I mentioned above, preferably conduct an experiment against them?

2. How does MIDUS-HML scale on 7B–13B backbones? Any obstacles (e.g., memory-bank thrashing) at larger widths?

3. Can you report latency/flops scaling with sequence length and batch size (train & inference), and compare to DUS? Where’s the crossover vs. dense FFN?

4. How sensitive are results to k, n, and the per-head transform size in HIVE? Any head-importance-aware allocation strategies tried?

5. For CPT, do MIDUS and DUS baselines share identical data order, schedule, and optimizer hyperparameters? If not, please provide tuned-per-method tables and/or a unified ablation.

6.

---

> ### Author Response · Authors · 2025-11-29
>
> Thank you for the thoughtful and detailed review. In the revision, we clarify our motivation and relation to MoD/DLO, add 8B-scale and additional SFT and math-reasoning experiments, extend latency and sensitivity analyses (sequence length, batch size, k and n), and make the shared training protocol across DUS and MIDUS–HML explicit. Below we respond to each of your points in detail.
>
> ---
> > W0) **Some claims, especially those related to the major motivation, that existing methods rely "on dense feed-forward networks", are not accurate. For example, papers [1],[2] are using "mixture of depth" like a sparse module for up-scaling. There is no discussion on the difference between these works, and they were not included as baselines.**
>
> Thank you for sharing these valuable references. We acknowledge that the works suggested by the reviewer are closely related to the literature on DUS efficiency, and we have incorporated their discussion and clarified their differences from our approach in the Introduction and Related Work sections. For further details, please refer to our response to Q1).
>
> ---
> > W1) **All results use a 1B backbone. Claims about general-purpose LLM scaling would be stronger with a 7B-class (or larger) model and at least one instruction-tuned setting beyond Alpaca-GPT-4.**
>
>
> We have added a full set of experiments on the LLaMA-3.1-8B backbone:
>
> - Table 2 now reports CPT and SFT results for LLaMA-3.1-8B across DUS baselines and MIDUS–HML. MIDUS–HML achieves the best average zero-shot accuracy and perplexity while using fewer parameters and less GPU memory than DUS, showing that our method scales favorably beyond the 1B regime. For efficiency of MIDUS-HML for LLaMA-3.1-8B, refer to the response to Q2)
> - **Table 2) CPT-Llama-3.1-8B on FineWeb-Edu Subset**
>
> | Method        | Wiki-PPL |   ARC | LogiQA |  Wino |  CSQA | BoolQ |  PIQA |  MMLU |   Average |
> | ------------- | -------: | ----: | -----: | ----: | ----: | ----: | ----: | ----: | --------: |
> | Base         |     8.35 | 79.97 |  26.88 | 72.06 | 65.19 | 81.83 | 78.84 | 58.61 |     66.20 |
> | SOLAR        |     9.90 | 79.88 |  26.88 | 71.59 | 57.41 | 80.70 | 78.56 | 54.37 |     64.20 |
> | LLaMA Pro    |     7.81 | 81.61 |  **29.49** | 73.72 | 70.93 | 81.65 | 79.98 | 62.56 |     68.56 |
> | LESA         |     7.73 | 82.07 |  27.96 | 74.11 | **72.40** | 81.93 | 80.30 | 62.63 |     68.77 |
> | OpT-DeUS     |     7.73 | 82.07 |  27.34 | **74.74** | 71.91 | 82.26 | **80.79** | 62.96 |     68.87 |
> | Avg-DeUS     |     7.95 | 82.15 |  27.50 | 73.48 | 71.09 | 82.17 | 80.20 | 62.11 |     68.39 |
> | **MIDUS-HML** | **7.40** | **82.37** |  28.57 | 74.59 | 70.84 | **82.87** | 80.25 | **63.40** | **68.98** |
>
> - **Table 2) SFT-Llama-3.1-8B with Alpaca-GPT-4 on FineWeb-Edu Subset Pretrained Mdoel**
>
> | Method        | Wiki-PPL |   ARC | LogiQA |  Wino |  CSQA | BoolQ |  PIQA |  MMLU |   Average |
> | ------------- | -------: | ----: | -----: | ----: | ----: | ----: | ----: | ----: | --------: |
> | Base         |     8.32 | 81.10 |  24.58 | 72.14 | 68.30 | 82.14 | 79.71 | 59.17 |     66.73 |
> | SOLAR        |     9.68 | 80.68 |  25.19 | 71.19 | 61.81 | 81.19 | 79.16 | 55.03 |     64.80 |
> | LLaMA Pro    |     7.81 | 83.33 |  27.19 | 74.11 | 72.07 | 82.26 | 80.79 | 62.32 |     68.87 |
> | LESA         |     7.72 | 83.84 |  26.57 | 75.53 | 73.05 | 83.00 | 80.69 | 63.57 |     69.47 |
> | OpT-DeUS     |     7.73 | 83.80 |  26.73 | **76.09** | 73.05 | 83.36 | 80.85 | 63.84 |     69.67 |
> | Avg-DeUS     |     7.91 | **83.88** |  26.42 | 75.45 | 72.89 | 83.18 | 80.47 | 63.10 |     69.34 |
> | **MIDUS-HML** | **7.50** | 83.50 |  **28.11** | 74.90 | **73.63** | **83.39** | **80.96** | **64.54** | **69.86** |

---

> > ### Author Response · Authors · 2025-11-29
> >
> > - For additional SFT experiments, Table 6 adds SFT results on Databricks-Dolly-15k [3], demonstrating that CPT gains from MIDUS–HML transfer robustly.
> >
> > - **Table 6) SFT-Llama-3.1-8B with Databricks-Dolly-15k on FineWeb-Edu Subset Pretrained Mdoel**
> >
> > | Methods       |  Wiki-PPL |       ARC |    LogiQA |      Wino |      CSQA |     BoolQ |      PIQA |      MMLU |   Average |
> > | ------------- | --------: | --------: | --------: | --------: | --------: | --------: | --------: | --------: | --------: |
> > | Base          |     13.07 |     68.64 |     21.35 |     60.30 |     27.52 |     63.27 | **75.24** |     30.49 |     49.55 |
> > | SOLAR         |     13.19 | **69.40** |     23.04 |     59.04 |     27.52 |     60.03 |     75.14 |     30.77 |     49.28 |
> > | LLaMA Pro     |     12.47 |     68.10 | **23.81** |     60.54 |     41.93 |     63.70 | **75.24** |     34.03 |     52.48 |
> > | LESA          | **11.77** |     66.46 |     23.66 |     60.77 | **49.06** |     64.98 |     74.86 |     38.22 |     54.00 |
> > | OpT-DeUS      |     11.78 |     67.21 | **23.81** |     61.48 | **49.14** |     63.76 |     75.19 |     37.83 |     54.06 |
> > | Avg-DeUS      |     12.10 |     67.51 |     22.73 |     60.30 |     46.60 |     65.02 |     74.37 |     36.79 |     53.33 |
> > | **MIDUS–HML** | **11.72** |     66.62 |     22.12 | **61.72** |     48.65 | **66.51** |     75.14 | **38.59** | **54.19** |
> >
> >
> > These additions directly address the concern about larger-scale and more diverse instruction-tuned settings
> >
> > ---
> > >W2) **Iteration times are reported, but sensitivity to sequence length (e.g., 8k–32k) isn’t analyzed; PKM’s two-stage top-k might have different break-even points. (Tables 2–4 give per-iter stats only)**
> >
> > We report latency as a function of both sequence length and batch size on Llama-3.2-1B and Llama-3.1-8B. Figure 3 and Figure 5 show prefill time as a function of prompt length, directly addressing the reviewer’s concern that PKM’s two stage Top k procedure may have different break even points at longer contexts. Figure 6 (Appendix D) varies the global batch size while fixing gradient accumulation steps and sequence length, and shows that MIDUS–HML consistently uses less GPU memory than DUS and achieves comparable or faster training time across tested batch sizes. Figure 7 (Appendix D) then varies sequence length. For Llama-3.1-8B, MIDUS–HML is strictly faster than all DUS baselines for all lengths. For Llama-3.2-1B, DUS can be slightly faster for very short sequences, but MIDUS–HML becomes faster as sequence length increases, including in the longer range highlighted by the reviewer. For more details, refer to the response to Q3)
> >
> > ---
> > >W3) **Helpful ablations are included, but further disentangling the impact of (i) Attn′ vs. full MHA, (ii) exact k and n choices, and (iii) HIVE’s parameterization per head would clarify where the gains come from.**
> >
> > Appendix F.4 (Table 10) now ablates the structure of HML, including:
> >
> > - Using the full MHA output vs. Attn′ as the query.
> > - Removing query normalization vs. using normalized queries.
> > - Adding or removing the HML residual connection.
> >
> > These experiments show that keeping Attn′ without normalization and retaining the HML residual yields the best trade-off, and that simply adding an extra FFN-like block (full MHA + FFN) does not match the gains of HML.
> >
> > - **Appendix F.4-Table 10) Ablation studies on Structure of HML Memory Layer**
> >
> > | Method               |  Wiki-PPL |       ARC |    LogiQA |      Wino |      CSQA |     BoolQ |      PIQA |      MMLU |   Average |
> > | -------------------- | --------: | --------: | --------: | --------: | --------: | --------: | --------: | --------: | --------: |
> > | **MIDUS–HML**        | **11.64** |     66.16 |     23.20 | **61.56** |     46.27 | **65.29** |     75.08 |     36.91 |     53.50 |
> > | w/ BatchNorm1D       |     11.74 | **66.54** |     22.43 |     60.77 | **47.34** |     64.80 |     74.92 |     36.94 |     53.39 |
> > | w/ LayerNorm         |     11.68 |     66.08 |     22.43 | **61.56** |     46.68 |     65.17 | **75.19** | **37.34** |     53.49 |
> > | w/ Residual          | **11.64** |     66.04 |     22.89 |     61.48 |     46.36 |     65.17 | **75.35** |     36.75 |     53.43 |
> > | w/ Output Projection | **11.64** |     66.12 | **23.66** |     61.09 |     46.52 |     65.23 |     74.86 |     37.07 | **53.51** |
> >
> > We refer the reviewer to our response to Q4) for the remaining discussion of the choices of k and n, as well as the per-head parameterization of HIVE.

---

> > > ### Author Response · Authors · 2025-11-29
> > >
> > > > W4) **The benchmark suite is knowledge-centric; evaluating on reasoning-heavy or long-context tasks would test whether HML helps beyond factual retrieval**
> > >
> > > We now include CPT experiment on math-domain reasoning, MathPile [4] and benchmarks, GSM8K, GSM8K-CoT, MATH, and MathQA (Sec. 5.2; Tables 5). These datasets require multi-step reasoning and mathematical problem solving, not just factual recall. CPT on the MathPile shows that MIDUS–HML achieves the best performance on almost all math benchmarks for both 1B and 8B backbones, indicating that head-wise memory improves challenging reasoning skills as well
> > >
> > > - **Table 5) CPT on MathPile — Llama-3.2-1B (5-shot accuracy)**
> > >
> > > | Method    | GSM8K | GSM8K-CoT | MATH | MathQA | Average |
> > > | --------- | :---: | :-------: | :--: | :----: | :-----: |
> > > | Base      |  4.32 |    4.93   | 5.66 |  29.88 |  14.93  |
> > > | LLaMA Pro |  5.61 |    4.40   | 5.64 |  30.62 |  15.42  |
> > > | OpT-DeUS  |  6.07 |    6.22   | **5.80** |  30.95 |  16.35  |
> > > | MIDUS-HML |  **6.60** |    **6.60**   | 5.62 |  **31.39** |  **16.73**  |
> > >
> > > - **Table 5)  CPT on MathPile — Llama-3.1-8B (5-shot accuracy)**
> > >
> > > | Method    | GSM8K | GSM8K-CoT |  MATH | MathQA | Average |
> > > | --------- | :---: | :-------: | :---: | :----: | :-----: |
> > > | Base      | 35.33 |   33.36   | 11.42 |  37.99 |  39.37  |
> > > | LLaMA Pro | 41.93 |   41.77   | 13.10 |  40.87 |  45.89  |
> > > | OpT-DeUS  | **49.36** |   50.04   | 13.84 |  42.58 |  51.94  |
> > > | MIDUS-HML | 48.90 |   **51.78**   | **14.34** |  **42.58** |  **52.53**  |
> > >
> > > ---
> > > > Q1) **Can you detail the difference/limitation of the related works I mentioned above, preferably conduct an experiment against them?**
> > >
> > > We have expanded Section. 2.2 to contrast MIDUS with Mixture of Depths [1] and DLO [2] in detail. These methods use dynamic routing over duplicated Transformer blocks, where each block still contains a full FFN and must be stored in memory. As a result, even though compute per token can be reduced at inference time, the parameter and activation-memory scaling remains tied to replicated FFNs. MIDUS instead moves additional capacity into compact head-wise memory banks, decoupling capacity growth from FFN size and enabling much lower parameter counts and training-time memory.
> > >
> > > We are focus on strong static DUS baselines that follow the same progressive up scaling protocol, and we show that MIDUS HML provides consistent gains in performance and efficiency over them at both 1B and 8B scales. We agree that incorporating MoD and DLO style methods as baselines would make the empirical study more complete and nuanced, and, if the paper is accepted, we plan to explore adding these comparisons in the camera ready version as time and compute budget permit.
> > >
> > > ---
> > > > Q2) **How does MIDUS-HML scale on 7B–13B backbones? Any obstacles (e.g., memory-bank thrashing) at larger widths?**
> > >
> > > The main practical issue we encountered was not an instability of HML on wider backbones, but a generic implementation bottleneck in the backward pass of the memory layer. When many tokens retrieve the same memory slot, a naive implementation based on `EmbeddingBag` or `scatter_add` issues many concurrent `atomicAdd` operations to the same address, which leads to severe atomic contention and an effective memory-thrashing behavior on the GPU [5]. This situation can arise on any backbone whenever the access distribution is sufficiently skewed.
> > >
> > > Appendix C.3 explains how we resolve this bottleneck with a custom backward kernel for the memory layer. The kernel first flattens all indices and token-wise gradients, then deduplicates indices within the batch, accumulates gradients per unique index in a local buffer, and finally applies a single aggregated update per slot to the global embedding gradients. This deduplication and pre-aggregation scheme is mathematically equivalent to the naive gradient, but it replaces many contending atomic writes with one write per slot and eliminates the memory-thrashing bottleneck while making better use of memory bandwidth.
> > >
> > > Beyond this backward-path optimization, Appendices C.1 and C.2 describe additional implementation-level improvements for more efficient inference, especially at shorter sequence lengths. Concretely, we replace the two-stage Top-k procedure in PKM with a mathematically equivalent single Top-k operation, and precompute and cache per–memory-head value banks at the start of inference to avoid repeated value-bank computation. Together, these optimizations yield a more efficient realization of MIDUS–HML.

---

> ### Author Response · Authors · 2025-11-29
>
> With this optimization in place we do not observe any critical obstacles when moving to larger backbones. In the revised paper we now include experiments on Llama-3.1-8B and MIDUS HML trains stably in this setting, Table 4, for example. The efficiency gains over DUS also become stronger at 8B than at 1B. Once the backward kernel in Appendix C.3 is used, MIDUS HML therefore becomes increasingly attractive as model width grows. The table below summarizes the efficiency comparison on Llama-3.1-8B.
>
> - **Table 4) Efficiency of DUS and MIDUS for Llama-3.1-8B**
>
> | Family | Method    | Params (trainable / total, B) | GPU memory (train / infer, GB) | Train time (s/iter) | Throughput (tokens/s) |
> |--------|-----------|--------------------------------|---------------------------------|----------------------|------------------------|
> | DUS    | LLaMA Pro | 3.49 / 11.52                   | 63.8 / 27.0                     | 6.89                 | 41.3                   |
> | DUS    | OpT-DeUS  | 3.49 / 11.52                   | 59.8 / 27.0                     | 5.78                 | 41.3                   |
> | MIDUS  | Linear    | 1.48 / 9.51                    | 51.3 / 23.1                     | 11.05                | 42.4                   |
> | MIDUS  | PKM       | 1.21 / 9.24                    | 48.3 / 22.5                     | 9.88                 | 47.3                   |
> | MIDUS  | HML       | **0.42 / 8.45**               | **34.8 / 20.1**                | **5.05**            | **50.5**              |
>
>
> ---
> > Q3) **Can you report latency/flops scaling with sequence length and batch size (train & inference), and compare to DUS? Where’s the crossover vs. dense FFN?**
>
> We provide additional latency analyses to characterize scaling on both Llama-3.2-1B and Llama-3.1-8B.
>
> - Figure 3 and 5 report prefill time as a function of prompt length.
> - Figure 6 (Appendix D) varies the global batch size while fixing gradient-accumulation steps and sequence length, showing that MIDUS–HML consistently uses less GPU memory than DUS and achieves comparable or faster training time across tested batch sizes.
> - Figure 7 (Appendix D) varies sequence length. For Llama-3.1-8B, MIDUS–HML is strictly faster than all DUS baselines for all lengths. For Llama-3.2-1B, DUS can be slightly faster for very short sequences, but MIDUS–HML becomes faster as sequence length increases.
>
> Overall, we observe a clear crossover: DUS can be marginally faster for smaller models and very short contexts, but as model size or sequence length increases, MIDUS–HML offers better or comparable latency while maintaining strictly lower memory usage, especially at the 8B scale. This behavior is robust across backbone size, sequence length, and batch size, where MIDUS–HML consistently requires less GPU memory.
>
> Appendix C further indicates that additional kernel-level optimizations of the memory layer (for example, fully fused HML kernels in Appendix C.3) and algorithmic improvements (such as a mathematically equivalent single Top-k operation and caching per–memory-head value banks in Appendices C.1 and C.2) are likely to shift this crossover further in favor of MIDUS–HML.

---

> > ### Author Response · Authors · 2025-11-29
> >
> > > Q4) **How sensitive are results to k, n, and the per-head transform size in HIVE? Any head-importance-aware allocation strategies tried?**
> >
> > **Sensitivity to k and n.**
> > - **Appendix F.1-Table 7) Sensitivity to the memory size n**
> >
> > |  # Memories |  Wiki-PPL |    ARC    |   LogiQA  |    Wino   |    CSQA   |   BoolQ   |   PIQA  |    MMLU   |  Average  |
> > | ----------: | :-------: | :-------: | :-------: | :-------: | :-------: | :-------: | :-----: | :-------: | :-------: |
> > |  66K (n=16) | **11.63** |  *66.12*  |   22.89   |   60.06   |  *46.27*  |   65.02   |  74.81  |  *36.74*  |   53.13   |
> > | 262K (n=32) |  *11.64*  |   65.95   | **23.35** |  *60.85*  | **47.17** | **65.44** |  74.76  |   36.41   |  *53.42*  |
> > |   1M (n=64) |  *11.64*  | **66.16** |  *23.20*  | **61.56** |  *46.27*  |  *65.29*  | *75.08* | **36.91** | **53.50** |
> >
> > The sensitivity of MIDUS–HML to the memory size n was already presented in the original submission. In the revised version, we reorganize these results in Appendix F.1 by varying n ∈ {16, 32, 64}, corresponding to roughly 66K, 262K, and 1M memories. As shown in the table above, average zero-shot accuracy increases from 53.13 (66K) to 53.42 (262K) and 53.50 (1M), while Wiki-PPL stays essentially flat in the range 11.63–11.64. We interpret this as evidence that larger tables gradually help tasks that benefit from retrieval, without affecting the core language modeling behavior.
> >
> > - **Appendix F.2-Table 8) Sensitivity to top-k**
> >
> > | top-k |  Wiki-PPL |    ARC    |   LogiQA  |    Wino   |    CSQA   |   BoolQ   |   PIQA  |    MMLU   |  Average  |
> > | ----: | :-------: | :-------: | :-------: | :-------: | :-------: | :-------: | :-----: | :-------: | :-------: |
> > |     1 |   11.67   |   65.74   |   21.97   |   60.54   |   46.19   |   64.95   |  74.86  |   36.95   |   53.03   |
> > |     2 | **11.63** |   65.87   | **23.20** |  *60.69*  |   46.11   |  *65.11*  |  74.54  |  *37.04*  |   53.22   |
> > |     4 |  *11.64*  | **66.16** | **23.20** | **61.56** |  *46.27*  | **65.29** | *75.08* |   36.91   | **53.50** |
> > |     8 |  *11.64*  |  *66.04*  |  *23.04*  |   60.46   | **46.93** |   64.98   | *75.03* | **37.11** |  *53.37*  |
> >
> > In addition, we newly introduce a sensitivity study for the number of retrieved memory per token, varying top-k ∈ {1, 2, 4, 8} in Appendix F.2. The corresponding table shows that average accuracy remains in a narrow band from 53.03 to 53.50 and perplexity again stays very stable.
> >
> > Overall, the small variation across both n and k suggests that MIDUS–HML is reasonably robust to these capacity and retrieval hyperparameters and does not rely on finely tuned top-k choices to perform well.
> >
> >
> > **Per-head transform size and HIVE.**
> > HML is structurally constrained so that each value in the value bank has dimension equal to the LLM hidden dimension divided by the number of heads. As a result, the per head transform size is also fixed to this value, and we could not vary it independently to conduct a separate sensitivity study.
> >
> > **Head-importance-aware allocation.**
> > We do not propose a head-importance-aware memory allocation scheme in this work. Instead, we argue that assigning a separate memory bank to each head already provides a sufficiently strong mechanism for enhancing head importance. In Section 5.3 and Figure 4, we analyze head importance before and after inserting HML and observe that HML amplifies the roles of already-important heads and increases the variance of head importance within each layer. Nevertheless, an explicit head-importance-aware allocation strategy (e.g., allocating additional capacity to more important heads) could further regulate head roles in MIDUS, and we leave this as a promising direction for future work.

---

> > > ### Author Response · Authors · 2025-11-29
> > >
> > > >Q5) **For CPT, do MIDUS and DUS baselines share identical data order, schedule, and optimizer hyperparameters? If not, please provide tuned-per-method tables and/or a unified ablation.**
> > >
> > >
> > > For CPT and SFT experiments we ensure that all DUS baselines and MIDUS variants share the same training setup. Section 4.1 and Appendix B now spell out the shared protocol. All methods are trained on the same FineWeb-Edu and MathPile subsets with identical tokenization, number of steps, warmup, cosine decay schedule, learning rate and weight decay under the fair comparison. We also fix the random seed for identical data order. For each backbone and dataset, we adopt the same global batch size, gradient-accumulation steps, and optimizer hyperparameters across all up-scaled models.
> > >
> > > To test the robustness of MIDUS–HML itself, Appendix F.3 (Table 9) varies the learning rate and weight decay for the memory-layer parameters while keeping the base training schedule fixed. MIDUS–HML achieves consistently strong performance across these settings,
> > >
> > > - **Appendix F.3-Table 9) Ablation studies on learning rate and weight decay**
> > >
> > > | LR   | WD    | Wiki-PPL |   ARC    | LogiQA  |  Wino   |  CSQA  | BoolQ  |  PIQA  | MMLU  | Average |
> > > |------|-------|----------|----------|---------|---------|--------|--------|--------|-------|---------|
> > > | 1e-4 | 1e-6  | 11.64    | 66.16    | **23.20** | **61.56** | 46.27  | 65.29  | **75.08** | 36.91 | **53.50** |
> > > | 1e-4 | 0     | 11.64    | *66.25*  | 21.97   | 60.77   | 46.60  | *65.32* | *75.03* | 36.99 | 53.27  |
> > > | 1e-4 | 1e-2  | **11.62** | 65.78   | *23.04* | 59.91   | 45.45  | **65.35** | 74.81 | **37.25** | 53.09  |
> > > | 1e-5 | 1e-6  | 11.64    | 65.57    | 21.51   | *61.01* | *46.85* | 63.58  | 74.48  | 37.12 | 52.87  |
> > > | 5e-5 | 1e-6  | *11.63*  | **66.46** | 22.89  | 60.14   | **47.58** | 64.86 | 74.54  | *37.23* | *53.39* |
> > >
> > > [1] Raposo, David, et al. "Mixture-of-depths: Dynamically allocating compute in transformer-based language models." arXiv preprint arXiv:2404.02258 (2024).
> > >
> > > [2] Tan Z, Dong D, Zhao X, et al. Dlo: Dynamic layer operation for efficient vertical scaling of llms[J]. arXiv preprint arXiv:2407.11030, 2024.
> > >
> > > [3] https://huggingface.co/datasets/databricks/databricks-dolly-15k
> > >
> > > [4] Wang, Zengzhi, et al. "Mathpile: A billion-token-scale pretraining corpus for math." Advances in Neural Information Processing Systems 37 (2024): 25426-25468.
> > >
> > > [5] https://github.com/pytorch/pytorch/issues/20655

---

> > > > ### Author Response · Authors · 2025-12-01
> > > > **Remind of Revisions Addressing Reviewer Concerns**
> > > >
> > > > We are grateful for this careful review. In response, we have refined the discussion of our motivation and the distinctions from Mixture-of-Depths and DLO, added 8B-scale CPT/SFT and math-reasoning experiments, broadened the latency and hyperparameter-sensitivity analyses, and made the shared training protocol across all DUS and MIDUS–HML variants explicit. We also remain fully available for any further questions or discussion about these revisions, if the review process allows.

---

### Official Review · Reviewer_9Rab · 2025-11-01

**Soundness:** 2
**Presentation:** 3
**Contribution:** 2
**Rating:** 4
**Confidence:** 2

**Summary:**

The paper proposes MIDUS, a novel method for scaling LLMs by increasing depth through memory-based rather than feed-forward expansion. MIDUS replaces these FFNs with Memory Blocks built around a Head-wise Memory Layer (HML), where each attention head maintains an independent memory bank for sparse retrieval. To further improve efficiency, the authors introduce Head-wise Implicit Value Expansion (HIVE), which factorizes per-head value spaces to preserve head alignment without redundant parameter storage. The design allows capacity to be added in a retrieval-based, head-aligned manner, effectively decoupling model quality gains from dense computation. Experiments on continual pre-training (CPT) and supervised fine-tuning (SFT) with Llama-3.2-1B demonstrate that MIDUS-HML consistently surpasses strong DUS baselines in both perplexity and zero-shot accuracy, while using fewer trainable parameters and less GPU memory.

**Strengths:**

- MIDUS achieves depth expansion through sparse retrieval rather than dense FFN projections, decoupling performance gains from the heavy parameter and activation costs of FFNs.
- MIDUS–HML achieves the low perplexity and high average zero-shot accuracy, particularly excelling in benchmarks such as CSQA, BoolQ, PIQA, and MMLU.
- HML assigns an independent memory bank per attention head, enabling selective retrieval aligned with head specialization, thereby minimizing cross-head interference compared to block-shared memories.

**Weaknesses:**

- The work lacks formal justification or theoretical analysis of why memory retrieval at head level leads to better generalization or gradient propagation.
- The paper would benefit from visualization or analysis of what the head-wise memories actually learn or retrieve—whether they store task-specific patterns, contextual cues, or token-level semantics.
- Since MIDUS replaces dense FFN expansion with sparse retrieval, it introduces the need to carefully determine memory size and placement, which may affect optimal scaling.

**Questions:**

- The authors fix the learning rate. Could the authors clarify whether this fixed rate was empirically tuned or simply adopted from earlier works?
- What underlying dynamics cause internal residual connections to weaken retrieval signals?
- What is the per-token inference overhead introduced by HML compared to standard FFNs?

---

> ### Author Response · Authors · 2025-11-29
>
> Thank you for the careful and constructive review. In the revision, we (i) add a formal head-importance analysis (Figure 4) to justify and visualize how head-wise memory changes gradient flow and specialization, and relate this to the observed generalization gains on FineWeb-Edu and MathPile, (ii) clarify memory-specific design choices, including capacity and placement ablations (Appendix F.1, F.5) and the fixed learning rate for keys/values, and (iii) analyze the effect of the internal residual in HML and quantify the per-token inference overhead of HML relative to DUS using our full efficiency tables and prefill-time curves for both 1B and 8B backbones.
>
> ---
> > **W1–W2) The work lacks formal justification or theoretical analysis of why memory retrieval at head level leads to better generalization or gradient propagation. The paper would benefit from visualization or analysis of what the head-wise memories actually learn or retrieve—whether they store task-specific patterns, contextual cues, or token-level semantics.**
>
> **Justification for head-wise memory**
>
> We agree that it is important to clarify why head-wise memory helps and to provide more direct evidence of what it changes inside the model. In the revision we therefore add a formal head-importance in Figure 4.
>
> To quantify the contribution of each attention head, we adopt the head-importance score $IS_h$ proposed by [1]. For a given head $h$, dataset $\mathcal{D}$, input prompt $x$, and answer $y$, let $a_h([x \mid y]) \in \mathbb{R}^{d_h}$ be the output of head $h$ on the concatenated sequence $[x \mid y]$, and $\mathcal{L}(y; x)$ be the negative log-likelihood loss on $y$ given $x$.
>
> The importance of head $h$ is defined as
>
> $$ IS_h(\mathcal{D})= \mathbb{E}_{(x,y)\sim\mathcal{D}} \left[a_h([x \mid y])^\top\frac{\partial \mathcal{L}(y;x)}{\partial a_h([x \mid y])}\right].
> $$
> The inner product is the directional derivative of the loss along the output direction of head $h$. It measures how much the loss would change if we infinitesimally scale the contribution of that head.
>
> We evaluate $IS_h$ on the PIQA benchmark in a zero-shot setting for two Llama-3.2-1B models:
>
> 1. **Base–Pretrained:** the original backbone before any up-scaling or CPT.
> 2. **MIDUS–HML:** the model with interleaved HML blocks after CPT on FineWeb-Edu.
>
> Both models share the same 16 Transformer layers in their backbone architecture. The Base model consists only of these 16 Transformer layers, whereas MIDUS–HML interleaves 8 additional memory (HML) blocks between them (resulting in 16 Transformer layers + 8 memory layers). During CPT with MIDUS–HML, all backbone parameters in the 16 Transformer layers are frozen and remain identical to the pretrained base weights, so that only the parameters in the 8 HML blocks are updated. When we compute the head-importance scores $IS_h$, we always measure them on the 16 Transformer layers that are common to both models. Therefore, any change in $IS_h$ between the Base and MIDUS–HML models must come from the presence and training of the interleaved HML blocks.

---

> > ### Author Response · Authors · 2025-11-29
> >
> > For each model we compute $IS_h$ for every head in every layer. Figure 4-a,b shows heatmaps of these scores, and Figure 4-c plots the per-layer variance of head importance. The MIDUS–HML heatmap in Figure 4-b is globally brighter than the Base–Pretrained heatmap in Figure 4-a, showing that head importance increases overall. Heads in the HML-augmented model have larger $IS_h$, meaning their outputs have stronger influence on the loss. Heads that were already important in the base model, bright spots in Figure 4-a, become significantly more important after inserting HML corresponding bright spots in Figure 4-b, while most other heads change only mildly. This indicates that the head-wise memory banks do not simply boost all heads. they preferentially amplify heads that are already useful for the task. Figure 4-c shows that the variance of $IS_h$ across heads within each layer increases under MIDUS–HML. The importance distribution becomes more skewed, with a small subset of heads carrying most of the importance in each layer. This suggests that head-wise memory leads to more specialized and concentrated head roles, instead of blurring them through block-shared FFNs.
> >
> > Together, these results provide a concrete view of what the head-wise memories learn. They create dedicated retrieval and gradient channels for each head, which strengthen the heads that the model already finds useful and sharpen their functional roles.
> >
> > **Relation to generalization**
> >
> > This sharpening of head roles is accompanied by consistent improvements in downstream performance. Across all of our CPT and SFT experiments on FineWeb-Edu (Tables 1–2) and math-domain CPT on MathPile (Table 5), MIDUS–HML achieves lower perplexity and higher average zero-shot accuracy than both the base backbone and strong DUS baselines, at both 1B and 8B scales. In particular, MIDUS–HML attains the best average 5-shot accuracy on GSM8K, GSM8K-CoT, MATH, and MathQA for both backbones, indicating that the head-wise memory banks are able to store and exploit domain-specific, high-level knowledge.
> >
> > We have added this head-importance analysis and the accompanying discussion of Figure 4 to the revised paper.
> >
> >
> > ---
> > >**W3) Since MIDUS replaces dense FFN expansion with sparse retrieval, it introduces the need to carefully determine memory size and placement, which may affect optimal scaling.**
> >
> > We appreciate this concern. While our original submission already included ablations on both memory size and DUS placement policy, in the revision we make these design choices more explicit and add further explanation and interpretation.
> >
> > - Table 1 varies the number of memories per head $n \in {16, 32, 64}$, corresponding to approximately 66K, 262K, and 1M composite memory slots. Average accuracy increases smoothly from 53.13 (66K) → 53.42 (262K) → 53.50 (1M), while Wiki-PPL remains essentially flat around 11.63–11.64. This suggests that HML benefits from larger capacity, but does not rely on a finely tuned value of (n) to perform well and  performance improves gradually rather than sharply peaking at a specific size.
> > - **Appendix F.1-Table 7) Sensitivity to the memory size n**
> >
> > |  # Memories |  Wiki-PPL |    ARC    |   LogiQA  |    Wino   |    CSQA   |   BoolQ   |   PIQA  |    MMLU   |  Average  |
> > | ----------: | :-------: | :-------: | :-------: | :-------: | :-------: | :-------: | :-----: | :-------: | :-------: |
> > |  66K (n=16) | **11.63** |  *66.12*  |   22.89   |   60.06   |  *46.27*  |   65.02   |  74.81  |  *36.74*  |   53.13   |
> > | 262K (n=32) |  *11.64*  |   65.95   | **23.35** |  *60.85*  | **47.17** | **65.44** |  74.76  |   36.41   |  *53.42*  |
> > |   1M (n=64) |  *11.64*  | **66.16** |  *23.20*  | **61.56** |  *46.27*  |  *65.29*  | *75.08* | **36.91** | **53.50** |
> >
> >
> > - We also already evaluated MIDUS–HML under *Top-heavy*, *Distributed*, and *Bottom-heavy* placement policies. In the revision, we highlight these results more clearly and discuss their implications (Appendix D). *Top-heavy* is most efficient in terms of memory and time, while Bottom-heavy slightly improves perplexity. The *Distributed* policy yields the best overall average zero-shot accuracy by injecting head-structured retrieval at multiple depths. Importantly, MIDUS–HML’s performance remains comparable to DUS baselines under all three policies, and efficiency results (Table 11 and Figures 6,7) show that MIDUS–HML can be configured to be both more accurate and more efficient than DUS once the placement policy is matched.
> > - **Appendix F.5-Table 11) Ablation study on the DUS placement policy**

---

> > > ### Author Response · Authors · 2025-11-29
> > >
> > > | DUS Policy     | GPU Memory (GB ↓) | Time (s/iter ↓) | Wiki-PPL ↓ |     ARC ↑ |  LogiQA ↑ |    Wino ↑ |    CSQA ↑ |   BoolQ ↑ |    PIQA ↑ |    MMLU ↑ | Average ↑ |
> > > | -------------- | ----------------: | --------------: | ---------: | --------: | --------: | --------: | --------: | --------: | --------: | --------: | --------: |
> > > | *Top-heavy*    |          **24.1** |        **3.89** |    *11.63* |   *66.12* |   *22.27* |   *61.25* | **46.44** |     64.28 |     74.92 |     36.23 |   *53.07* |
> > > | *Distributed*  |            *28.0* |          *4.56* |      11.64 | **66.16** | **23.20** | **61.56** |   *46.27* | **65.29** |   *75.08* | **36.91** | **53.50** |
> > > | *Bottom-heavy* |              28.6 |            4.67 |    *11.60* | **66.12** |     21.81 |     60.30 |     45.05 |   *65.05* | **75.30** |   *36.35* |     52.85 |
> > >
> > > ---
> > > >**Q1) The authors fix the learning rate. Could the authors clarify whether this fixed rate was empirically tuned or simply adopted from earlier works?**
> > >
> > > For CPT and SFT, all backbone and DUS parameters use the same optimizer hyperparameters and cosine schedule as in prior DUS work [2]. We did not introduce a different schedule there. The only exception is the MIDUS memory layer, where we decouple the learning rate of keys/values from the cosine schedule and keep it constant over training. We also apply no weight decay on key-value optimizer. This design is directly inspired by earlier work on PKM [3]. In the official PKM implementation [4], the value embeddings are trained with a fixed learning rate rather than a decayed schedule. We follow the same principle for both keys and values in HML.
> > >
> > > The reason is that, unlike standard dense parameters, memory slots are updated very sparsely due to top-k retrieval. At each step, only the keys/values corresponding to the retrieved slots receive gradient updates. Under a cosine schedule, slots that happen to be selected early in training would be updated with a much larger effective learning rate than slots that are selected later, purely due to timing rather than usefulness. This is undesirable. Once a key–value pair is referenced, we want it to be updated under the same learning-rate regime, regardless of when in training it starts being used.
> > >
> > > ---
> > > > **Q2) What underlying dynamics cause internal residual connections to weaken retrieval signals?**
> > >
> > > In our setting, the “internal residual” refers to modifying the query to the memory from
> > > $a' = \text{Attn}'(\mathrm{LN}(x))$ to $a'_{\text{res}} = x + \text{Attn}'(\mathrm{LN}(x))$, where $x$ is the output of the previous Transformer block. In the original Transformer blocks, the residual passing attention layer is learned jointly with an output projection $W_o$, so the model is optimized to add $W_o \cdot \text{MHA}(\mathrm{LN}(x))$ back to $x$. By contrast, in MIDUS–HML we deliberately remove this output projection and want the per-head outputs of $\text{Attn}'$ to serve directly as queries to the memory. Simply adding $x$ back at the query level does not restore the pretrained mechanism, instead it changes the object that drives retrieval into “raw block input + head output,” which the original model has never been trained to treat as a meaningful residual unit.
> > >
> > > Conceptually, the memory block already introduces a residual at the block level, $M(x) = x + m(x)$, and this is where we want the model to decide how much to trust the retrieved information. Inside the block, our design goal is different. Each head’s $\text{Attn}'*h$ should represent the "context as understood by that head", and the corresponding memory bank should store “what extra information is useful given this head-specific context.” Using $a'_{\text{res},h} = x + \text{Attn}'_h$ as the query reintroduces the block input $x$ into every head’s query, so retrieval is driven by something closer to a generic layer input than by a purely head-specific context representation. This weakens the differentiation between heads and goes against the goal of HML as a head-conditioned retrieval mechanism, especially since a large residual path $x + m(x)$ is already present at the memory-block output.
> > >
> > >
> > > Empirically, Appendix F.4 (Table 10) compares the default HML (no internal residual on the query) with a variant “w/ Residual” that uses $a'_{\text{res}}$. The residual variant achieves similar perplexity but worse average zero-shot accuracy, indicating that it does not provide a useful inductive bias on top of the existing block-level residual and, in practice, weakens the effectiveness of the head-wise memory retrieval. For this reason we adopt the simpler design that uses $\text{Attn}'$ directly as the query and keeps the residual structure only at the memory-block output.

---

> ### Author Response · Authors · 2025-11-29
>
> ---
> > **Q3) What is the per-token inference overhead introduced by HML compared to standard FFNs?**
>
> * For Llama-3.2-1B, Table 12 reports efficiency metrics and shows that MIDUS–HML achieves comparable generation throughput to DUS baselines, while using less GPU memory. Figure 5 plots "prefill time vs. prompt length", and indicates that DUS can be slightly faster at very short sequences, but MIDUS–HML catches up and becomes faster than DUS as the context length increases, leading to negative Δ Prefill Time in the longer-context regime.
>
> - **Table 3) Efficiency of DUS and MIDUS for Llama-3.2-1B**
>
> **Table: Efficiency of DUS and MIDUS for `Llama-3.2-1B` (trainable/total parameters, GPU memory for train/inference, training time, and generation throughput).**
>
> | Family | Method    | Params (trainable / total, B) | GPU memory (train / infer, GB) | Train time (s/iter) | Throughput (tokens/s) |
> | ------ | --------- | --------------- | ----------------- | ---------------- | ----------------------- |
> | DUS    | LLaMA Pro | 0.49 / 1.72     | 42.8 / 14.5       | 5.26             | **99.1**                |
> | DUS    | SOLAR     | 1.72 / 1.72     | 49.0 / 14.5       | 6.48             | **99.1**                |
> | DUS    | OpT-DeUS  | 0.49 / 1.72     | *38.3* / 14.5     | *4.58*           | **99.1**                |
> | MIDUS  | Linear    | 0.35 / 1.59     | 46.4 / 14.2       | 10.79            | 79.5                    |
> | MIDUS  | PKM       | *0.19 / 1.42*   | 43.2 / *14.0*     | 8.42             | 91.5                    |
> | MIDUS  | HML       | **0.05 / 1.29** | **30.1 / 13.5**   | **4.56**         | *98.6*                  |
>
>
> * For Llama-3.1-8B, Table 4 shows that MIDUS–HML achieves higher generation throughput than DUS, together with strictly lower training and inference memory. Figure 3 (prefill time vs. prompt length) further shows that MIDUS–HML has consistently negative Δ Prefill Time relative to DUS across the evaluated prompt lengths, i.e., it is strictly faster in the prefill phase at 8B.
>
> - **Table 4) Efficiency of DUS and MIDUS for Llama-3.1-8B**
>
> | Family | Method    | Params (trainable / total, B) | GPU memory (train / infer, GB) | Train time (s/iter) | Throughput (tokens/s) |
> |--------|-----------|--------------------------------|---------------------------------|----------------------|------------------------|
> | DUS    | LLaMA Pro | 3.49 / 11.52                   | 63.8 / 27.0                     | 6.89                 | 41.3                   |
> | DUS    | OpT-DeUS  | 3.49 / 11.52                   | 59.8 / 27.0                     | 5.78                 | 41.3                   |
> | MIDUS  | Linear    | 1.48 / 9.51                    | 51.3 / 23.1                     | 11.05                | 42.4                   |
> | MIDUS  | PKM       | 1.21 / 9.24                    | 48.3 / 22.5                     | 9.88                 | 47.3                   |
> | MIDUS  | HML       | **0.42 / 8.45**               | **34.8 / 20.1**                | **5.05**            | **50.5**              |
>
>
> Across both Llama-3.2-1B and Llama-3.1-8B, the end-to-end inference latency is comparable to or better than DUS for the context lengths we study, while providing better accuracy and lower memory usage.
>
>
> [1] Bansal, Hritik, et al. "Rethinking the role of scale for in-context learning: An interpretability-based case study at 66 billion scale." Proceedings of the 61st Annual Meeting of the Association for Computational Linguistics (Volume 1: Long Papers). 2023.
>
> [2] Cao, Mingzi, Xi Wang, and Nikolaos Aletras. "Progressive Depth Up-scaling via Optimal Transport." arXiv preprint arXiv:2508.08011 (2025).
>
> [3] Lample, Guillaume, et al. "Large memory layers with product keys." Advances in Neural Information Processing Systems 32 (2019).
>
> [4] https://github.com/facebookresearch/XLM/blob/main/PKM-layer.ipynb

---

> > ### Author Response · Authors · 2025-12-01
> > **Remind of Revisions Addressing Reviewer Concerns**
> >
> > We are grateful for this careful review. In the revision, we introduce a formal head-importance analysis (Figure 4) to justify and visualize how head-wise memory affects gradient flow and specialization, make our memory-specific design choices more explicit through capacity and placement ablations (Appendix F.1, F.5), clarify the rationale behind the fixed learning rate for keys and values, and quantify the per-token inference overhead of HML relative to DUS using the full efficiency tables and prefill-time curves at both 1B and 8B scales. We also remain fully available for any further questions or discussion about these revisions, if the review process allows.

---

### Official Review · Reviewer_xNN6 · 2025-11-01

**Soundness:** 2
**Presentation:** 2
**Contribution:** 2
**Rating:** 2
**Confidence:** 3

**Summary:**

This paper investigates depth upscaling to enhance LLM performance with light continued pretraining or supervised fine-tuning. Since attention heads specialize differently and additional dense FFN layers are computationally heavy, MIDUS assigns an independent memory bank to each head, enabling head-wise retrieval also in FFN. In experiments, the depth-upscaling layers adapt quickly with only light continued pretraining or supervised fine-tuning and deliver better accuracy than baseline models on commonsense reasoning tasks. The method consistently outperforms prior work while using fewer parameters and achieving higher efficiency.

**Strengths:**

* The paper proposes a memory-based alternative to FFN replication, motivated by the head-independent representations in attention. Thus, it can be more efficient than prior depth-scaling approaches that use dense FFN layers, and it may be easier to learn due to the sparse, head-wise connections.

* The experiments report both accuracy and efficiency. It also appears that the paper compares fairly with prior work and consistently outperforms it.

* The method is easy to understand, and the presentation is clear.

**Weaknesses:**

* Efficiency. As I understand it, the method adds additional layers. Then, why does it use fewer parameters and less GPU memory compared to the original model? Also, for latency, does the paper measure prefilling time or decoding time?

* Task coverage. The experiments seem to focus on general-purpose commonsense reasoning. Could the authors also report results on harder domains such as code and math? Baselines may already perform well in these specialized areas, whereas depth upscaling trained on 2B web tokens may not transfer as effectively. In short, can depth upscaling still perform well (better performance than the original model) on code, math, or long-context tasks under the 2B web token CPT setup?

* Comparisons. Why is the same zero-shot accuracy repeated in Table 2 as in Table 1? How does depth upscaling with CPT/SFT compare to training the same total number of layers from scratch? How does the method perform—in both efficiency and accuracy—on larger models such as 3B or 7B?


* MIDUS layer design. In MIDUS, there appears to be no hidden-state mixing across heads in the feedforward layer; the whole hidden-state mixing happens only in the initial projections that produce Q/K/V. Do the authors think this could introduce any implicit limitations?

**Questions:**

Please see above

---

> ### Author Response · Authors · 2025-11-29
>
> Thank you for the thoughtful and detailed review. In the revision, we (i) clarify that all efficiency claims are made in the depth up-scaling setting relative to DUS baselines (and make our prefill vs. decoding latency metrics explicit), (ii) add math-domain CPT experiments on MathPile and 8B-scale CPT/SFT results to broaden task coverage and demonstrate scalability, (iii) restructure the result tables to remove duplicated zero-shot numbers and isolate efficiency metrics, and (iv) expand the discussion and ablations around the MIDUS–HML layer design, showing how head-wise memory affects head importance and why the lack of intra-memory head mixing does not appear to be a practical limitation. Below we respond to each of your points in detail.
>
> ---
> >W1) **Efficiency. As I understand it, the method adds additional layers. Then, why does it use fewer parameters and less GPU memory compared to the original model? Also, for latency, does the paper measure prefilling time or decoding time?**
>
> We apologize for the confusion and clarify that all efficiency claims in the paper are made in the up-scaling setting, relative to DUS-style depth-upscaling methods, not relative to the pre-upscaled base model.
>
> We do not claim that MIDUS–HML has fewer total parameters than the original backbone. Instead, we compare against DUS baselines that add full FFN blocks on the same backbone. Under this matched up-scaling protocol, MIDUS–HML introduces substantially fewer additional parameters than DUS. For example, on Llama-3.1-8B (Table 4), the total parameter count is 11.52B for LLaMA Pro / OpT-DeUS versus 8.45B for MIDUS–HML, compared to 8.03B for the base model. Thus, MIDUS–HML is only slightly larger than the base model but significantly smaller than DUS at the same up-scaled depth.
> - **Table 4) Efficiency of DUS and MIDUS for Llama-3.1-8B**
>
> | Family | Method    | Params (trainable / total, B) | GPU memory (train / infer, GB) | Train time (s/iter) | Throughput (tokens/s) |
> |--------|-----------|--------------------------------|---------------------------------|----------------------|------------------------|
> | DUS    | LLaMA Pro | 3.49 / 11.52                   | 63.8 / 27.0                     | 6.89                 | 41.3                   |
> | DUS    | OpT-DeUS  | 3.49 / 11.52                   | 59.8 / 27.0                     | 5.78                 | 41.3                   |
> | MIDUS  | Linear    | 1.48 / 9.51                    | 51.3 / 23.1                     | 11.05                | 42.4                   |
> | MIDUS  | PKM       | 1.21 / 9.24                    | 48.3 / 22.5                     | 9.88                 | 47.3                   |
> | MIDUS  | HML       | **0.42 / 8.45**               | **34.8 / 20.1**                | **5.05**            | **50.5**              |
>
> In our CPT/SFT setup we train only the added depth-upscaling blocks. Consequently, the number of trainable parameters is also much smaller for MIDUS–HML or DUS than for original Base model (e.g., 0.42B vs. 8.03B at 8B scale), while the total number of parameters and peak GPU memory at inference stage of MIDUS-HML or DUS are still larger than the base model.
>
> Compared with DUS, we assign the same number of expanded block for both DUS and MIDUS. Since HML blocks are lightweight than Transformer blocks, the peak training GPU memory for MIDUS–HML is lower than the DUS baselines at the same up-scaled depth (Tables 4 and 12). This is the main regime where we emphasize memory savings.
>
> We now make explicit that we measure both prefill latency and decoding throughput.
> We report as “∆ Prefill Time” relative to DUS in Figures 3 and 5 (negative values indicate faster prefill than DUS).
> For decoding throughput, we report as “Throughput (tokens/s)” in Table 4 and Table 12 in the Appendix, which corresponds to the decoding phase after prefill.
>
> Empirically, MIDUS–HML matches or slightly lags DUS on prefill time for very short sequences, but becomes faster as sequence length or model size increases, and is strictly faster than all DUS baselines on Llama-3.1-8B across the lengths we test. For decoding, MIDUS–HML achieves comparable throughput to DUS at the 1B scale and the highest throughput among all up-scaled models at the 8B scale, while still using less GPU memory than DUS.

---

> > ### Author Response · Authors · 2025-11-29
> >
> > ---
> > >**W2) Task coverage. The experiments seem to focus on general-purpose commonsense reasoning. Could the authors also report results on harder domains such as code and math? Baselines may already perform well in these specialized areas, whereas depth upscaling trained on 2B web tokens may not transfer as effectively. In short, can depth upscaling still perform well (better performance than the original model) on code, math, or long-context tasks under the 2B web token CPT setup?**
> >
> > We agree that testing beyond general-purpose reasoning is important. In the revision,  we therefore added math-domain CPT experiments on MathPile (1.1B) [1], targeting tasks that require multi-step reasoning. For more details of dataset and experiment setting, refer to Appendix B.
> >
> >
> > - **Table 5) CPT on MathPile — Llama-3.2-1B (5-shot accuracy)**
> >
> > | Method    | GSM8K | GSM8K-CoT | MATH | MathQA | Average |
> > | --------- | :---: | :-------: | :--: | :----: | :-----: |
> > | Base      |  4.32 |    4.93   | 5.66 |  29.88 |  14.93  |
> > | LLaMA Pro |  5.61 |    4.40   | 5.64 |  30.62 |  15.42  |
> > | OpT-DeUS  |  6.07 |    6.22   | **5.80** |  30.95 |  16.35  |
> > | MIDUS-HML |  **6.60** |    **6.60**   | 5.62 |  **31.39** |  **16.73**  |
> >
> > - **Table 5)  CPT on MathPile — Llama-3.1-8B (5-shot accuracy)**
> >
> > | Method    | GSM8K | GSM8K-CoT |  MATH | MathQA | Average |
> > | --------- | :---: | :-------: | :---: | :----: | :-----: |
> > | Base      | 35.33 |   33.36   | 11.42 |  37.99 |  39.37  |
> > | LLaMA Pro | 41.93 |   41.77   | 13.10 |  40.87 |  45.89  |
> > | OpT-DeUS  | **49.36** |   50.04   | 13.84 |  42.58 |  51.94  |
> > | MIDUS-HML | 48.90 |   **51.78**   | **14.34** |  **42.58** |  **52.53**  |
> >
> >
> > Section 5.2 and Table 5 report CPT results on MathPile evaluated on GSM8K, GSM8K–CoT, MATH, and MathQA for both Llama-3.2-1B and Llama-3.1-8B. MIDUS–HML achieves the best average 5-shot accuracy on all math benchmarks and both backbones. These tasks are substantially more reasoning-heavy than our knowledge-centric suite and indicate that the depth-upscaled memory blocks can store and exploit specialized mathematical knowledge.
> >
> >  ---
> > >W3) **Comparisons. Why is the same zero-shot accuracy repeated in Table 2 as in Table 1? How does depth upscaling with CPT/SFT compare to training the same total number of layers from scratch? How does the method perform—in both efficiency and accuracy—on larger models such as 3B or 7B?**
> >
> > We thank the reviewer for pointing out that zero-shot accuracy numbers were repeated between Tables 1 and 2 in our original submission. Our original intention was to place performance and efficiency side by side to facilitate comparison, which led us to duplicate the accuracy entries in Table 2. We agree that this layout is redundant. In the revision, we remove the repeated numbers and instead restructure the results. Table 1 now reports CPT and SFT performance across DUS baselines and MIDUS–HML on Llama-3.2-1B, Table 3 compares different MIDUS variants (e.g., Linear, PKM, HML), and Table 12 in the Appendix collects the efficiency metrics (parameter counts, peak training/inference GPU memory, training time, and generation throughput), so that no table contains duplicated values.
> >
> > Regarding the comparison to “training the same total depth from scratch,” we do not re-pretrain a deeper model from scratch, but our setup already includes a closely related (and arguably more favorable) condition for the base model. At 8B scale the base model has 32 Transformer blocks (16 at 1B scale), and under CPT it updates all of these layers. In contrast, DUS and MIDUS–HML add 16 blocks at 8B (8 at 1B) and train only these newly inserted blocks while freezing the original backbone. Despite this advantage in the number of trainable layers and total updated parameters, the base model consistently underperforms both DUS and MIDUS–HML across all our experiments: FineWeb-Edu CPT and SFT at 1B and 8B (Table 1 and 2), as well as MathPile CPT at 1B and 8B (Table 5).

---

> ### Author Response · Authors · 2025-11-29
>
> - **Table 2) CPT-Llama-3.1-8B on FineWeb-Edu Subset**
>
> | Method        | Wiki-PPL |   ARC | LogiQA |  Wino |  CSQA | BoolQ |  PIQA |  MMLU |   Average |
> | ------------- | -------: | ----: | -----: | ----: | ----: | ----: | ----: | ----: | --------: |
> | Base         |     8.35 | 79.97 |  26.88 | 72.06 | 65.19 | 81.83 | 78.84 | 58.61 |     66.20 |
> | SOLAR        |     9.90 | 79.88 |  26.88 | 71.59 | 57.41 | 80.70 | 78.56 | 54.37 |     64.20 |
> | LLaMA Pro    |     7.81 | 81.61 |  **29.49** | 73.72 | 70.93 | 81.65 | 79.98 | 62.56 |     68.56 |
> | LESA         |     7.73 | 82.07 |  27.96 | 74.11 | **72.40** | 81.93 | 80.30 | 62.63 |     68.77 |
> | OpT-DeUS     |     7.73 | 82.07 |  27.34 | **74.74** | 71.91 | 82.26 | **80.79** | 62.96 |     68.87 |
> | Avg-DeUS     |     7.95 | 82.15 |  27.50 | 73.48 | 71.09 | 82.17 | 80.20 | 62.11 |     68.39 |
> | **MIDUS-HML** | **7.40** | **82.37** |  28.57 | 74.59 | 70.84 | **82.87** | 80.25 | **63.40** | **68.98** |
>
> - **Table 2) SFT-Llama-3.1-8B with Alpaca-GPT-4 on FineWeb-Edu Subset Pretrained Mdoel**
>
> | Method        | Wiki-PPL |   ARC | LogiQA |  Wino |  CSQA | BoolQ |  PIQA |  MMLU |   Average |
> | ------------- | -------: | ----: | -----: | ----: | ----: | ----: | ----: | ----: | --------: |
> | Base         |     8.32 | 81.10 |  24.58 | 72.14 | 68.30 | 82.14 | 79.71 | 59.17 |     66.73 |
> | SOLAR        |     9.68 | 80.68 |  25.19 | 71.19 | 61.81 | 81.19 | 79.16 | 55.03 |     64.80 |
> | LLaMA Pro    |     7.81 | 83.33 |  27.19 | 74.11 | 72.07 | 82.26 | 80.79 | 62.32 |     68.87 |
> | LESA         |     7.72 | 83.84 |  26.57 | 75.53 | 73.05 | 83.00 | 80.69 | 63.57 |     69.47 |
> | OpT-DeUS     |     7.73 | 83.80 |  26.73 | **76.09** | 73.05 | 83.36 | 80.85 | 63.84 |     69.67 |
> | Avg-DeUS     |     7.91 | **83.88** |  26.42 | 75.45 | 72.89 | 83.18 | 80.47 | 63.10 |     69.34 |
> | **MIDUS-HML** | **7.50** | 83.50 |  **28.11** | 74.90 | **73.63** | **83.39** | **80.96** | **64.54** | **69.86** |
>
> To demonstrate that MIDUS–HML remains effective and efficient beyond the 1B regime, we additionally run experiments on an 8B backbone. On Llama-3.1-8B, we report CPT and SFT results on the FineWeb-Edu subset (Table 2) and CPT results on MathPile (Table 5), showing that MIDUS–HML achieves the best average performance among up-scaled models. For efficiency, Table 4 summarizes parameter counts, peak training/inference memory, training time, and generation throughput. In addition, Figure 3 reports prefill time as a function of prompt length, and Figures 6 and 7 vary batch size and sequence length for training-time latency. These results indicate that MIDUS–HML is not only competitive at 1B but becomes more efficient relative to DUS at 8B, offering stronger performance gains with fewer additional parameters, lower training memory, and comparative or better latency.
> - **Table 4) Efficiency of DUS and MIDUS for Llama-3.1-8B**
>
> | Family | Method    | Params (trainable / total, B) | GPU memory (train / infer, GB) | Train time (s/iter) | Throughput (tokens/s) |
> |--------|-----------|--------------------------------|---------------------------------|----------------------|------------------------|
> | DUS    | LLaMA Pro | 3.49 / 11.52                   | 63.8 / 27.0                     | 6.89                 | 41.3                   |
> | DUS    | OpT-DeUS  | 3.49 / 11.52                   | 59.8 / 27.0                     | 5.78                 | 41.3                   |
> | MIDUS  | Linear    | 1.48 / 9.51                    | 51.3 / 23.1                     | 11.05                | 42.4                   |
> | MIDUS  | PKM       | 1.21 / 9.24                    | 48.3 / 22.5                     | 9.88                 | 47.3                   |
> | MIDUS  | HML       | **0.42 / 8.45**               | **34.8 / 20.1**                | **5.05**            | **50.5**              |

---

> > ### Author Response · Authors · 2025-11-29
> >
> > >W4) **MIDUS layer design. In MIDUS, there appears to be no hidden-state mixing across heads in the feedforward layer; the whole hidden-state mixing happens only in the initial projections that produce Q/K/V. Do the authors think this could introduce any implicit limitations?**
> >
> > We appreciate this architectural question and have clarified the discussion in Section 4.2 and Appendix F.4. Within the HML block, we intentionally keep the memory head-wise. Each head has its own PKM key space and its own HIVE transform. This design is meant to amplify and refine the roles of individual heads rather than mix them immediately.
> >
> > In practice, this head-independent, no–hidden-state–mixing structure appears to be beneficial rather than limiting. As shown in Figure 4, inserting HML systematically increases overall head importance and, in particular, further amplifies already-important heads, suggesting that HML sharpens functional distinctions across heads instead of blurring them. Quantitatively, this effect is also reflected indirectly in Tables 3 and 10. Table 3 compares MIDUS variants and shows that Linear and PKM memories, which still apply an output projection after attention and thus mix hidden states across heads, do not match the performance of HML. Moreover, in Table 10, the variant that keeps an explicit output projection inside HML—thereby reintroducing cross-head mixing and increasing the total parameter count—performs very similarly to the original HML design. Taken together, these results suggest that effective head-wise memory does not require additional mixing inside the memory block; if anything, preserving strong per-head independence while leaving cross-head interaction to the surrounding attention and FFN layers is a more favorable design choice.
> >
> > - **Table 3) CPT results on FineWeb-Edu with Llama-3.2-1B for MIDUS variants.**
> >
> > | Memory Layer | Wiki-PPL ↓ |     ARC ↑ |  LogiQA ↑ |    Wino ↑ |    CSQA ↑ |   BoolQ ↑ |    PIQA ↑ |    MMLU ↑ | Average ↑ |
> > | ------------ | ---------: | --------: | --------: | --------: | --------: | --------: | --------: | --------: | --------: |
> > | Linear       |    *11.82* |     65.61 |   *22.73* | **62.04** |   *42.42* |     64.71 | **75.08** |   *35.89* |   *52.64* |
> > | PKM          |      11.97 |   *65.70* |     21.35 |     61.40 |     42.18 | **65.47** |   *74.65* |     33.93 |     52.10 |
> > | HML          |  **11.64** | **66.16** | **23.20** |   *61.56* | **46.27** |   *65.29* | **75.08** | **36.91** | **53.50** |
> >
> >
> > - **Appendix F.4-Table 10) Ablation studies on Structure of HML Memory Layer**
> >
> > | Method               |  Wiki-PPL |       ARC |    LogiQA |      Wino |      CSQA |     BoolQ |      PIQA |      MMLU |   Average |
> > | -------------------- | --------: | --------: | --------: | --------: | --------: | --------: | --------: | --------: | --------: |
> > | **MIDUS–HML**        | **11.64** |     66.16 |     23.20 | **61.56** |     46.27 | **65.29** |     75.08 |     36.91 |     53.50 |
> > | w/ BatchNorm1D       |     11.74 | **66.54** |     22.43 |     60.77 | **47.34** |     64.80 |     74.92 |     36.94 |     53.39 |
> > | w/ LayerNorm         |     11.68 |     66.08 |     22.43 | **61.56** |     46.68 |     65.17 | **75.19** | **37.34** |     53.49 |
> > | w/ Residual          | **11.64** |     66.04 |     22.89 |     61.48 |     46.36 |     65.17 | **75.35** |     36.75 |     53.43 |
> > | w/ Output Projection | **11.64** |     66.12 | **23.66** |     61.09 |     46.52 |     65.23 |     74.86 |     37.07 | **53.51** |
> >
> >
> > [1] Wang, Zengzhi, et al. "Mathpile: A billion-token-scale pretraining corpus for math." Advances in Neural Information Processing Systems 37 (2024): 25426-25468.

---

> ### Author Response · Authors · 2025-12-01
> **Remind of Revisions Addressing Reviewer Concerns**
>
> We thank the reviewer again for their detailed review. The added efficiency analyses, broader math and 8B-scale experiments, and the expanded discussion and ablations of the MIDUS–HML layer design directly address the issues you raised. We would also very much welcome any further discussion or questions about these revisions, should the review process allow.

---

### Author Response · Authors · 2025-11-29
**Global Response**

We thank all reviewers (`xNN6`, `9Rab`, `9rn7`, `NZXf`) for their constructive feedback. Below we summarize the main strengths and shared concerns, and how the revision addresses them. Reviewer-specific details appear in the individual responses.
Below, we summarize how the revision addresses key concerns. All changes in the revised manuscript are highlighted in blue text.

### **Strengths highlighted by reviewers**
* **[`9Rab`, `9rn7`, `NZXf`] Novel & Modular Design:** Praised for replacing dense FFNs with Head-wise Memory Layers (HML) and HIVE as an ingenious retrieval-based scaling approach.
* **[`xNN6`, `9Rab`, `NZXf`] Parameter & Memory Efficiency:** Highlighted strong performance using significantly fewer parameters and less GPU memory than DUS.
* **[`xNN6`, `9rn7`] Robust Performance:** Noted consistent outperformance in perplexity and zero-shot accuracy across diverse benchmarks.
* **[`xNN6`, `9rn7`] Clarity & Reproducibility:** Appreciated the formalization, diagrams, and reproducible code.

### **Concerns and how we addressed them**

**1. Efficiency & Latency: Metrics and Scaling [`xNN6`, `9rn7`, `9Rab`]**
* We clarified efficiency claims are relative to **up-scaled DUS baselines**. **Tables 4 & 12** now separate parameters, memory, and throughput. Results confirm MIDUS-HML uses **strictly less GPU memory** across all settings. In terms of generation throughput, MIDUS is **superior on 8B** and **comparable on 1B**. **Figures 3, 5, 6, and 7** demonstrate that MIDUS latency outperforms or matches DUS as sequence length increases. **Appendix C** details kernel optimizations (deduplicated aggregation, single Top-$k$, cached values) that resolve atomic contention and enable this scaling.

**2. Task Coverage & Scaling to Larger Models [`xNN6`, `9rn7`]**
* We added **Llama-3.1-8B** experiments. MIDUS-HML achieves the best zero-shot accuracy and perplexity among up-scaled models. We expanded to **MathPile CPT** and **Dolly-15k SFT**, where MIDUS **achieves superior average performance** compared to DUS baselines.

**3. Theoretical Justification & Design [`9Rab`, `xNN6`, `9rn7`]**
* A new **Head Importance** analysis (Fig. 4) visualizes that HML selectively amplifies and sharpens the roles of important heads. We corrected **Figures 1 & 2** to remove the internal residual; expanded ablations (Table 10) confirm that using **Attn′ as the query** without internal residuals offers the optimal efficiency-accuracy trade-off.

**4. Hyperparameters & Policy Sensitivity [`NZXf`, `9Rab`, `9rn7`]**
* **Appendices F.1–F.3** show performance is smooth across memory sizes ($n$) and Top-$k$. **Appendix F.5** confirms MIDUS remains competitive under **Top-heavy, Distributed, and Bottom-heavy** policies. **Appendix B** clarifies that all baselines share identical training protocols, with the exception of **fixed memory LR/no-weight-decay** for HML.

**5. Distinction from Dynamic/Sparse Methods [`9rn7`]**
* We expanded **Section 2.2** to contrast MIDUS with dynamic routing methods (MoD, DLO). Unlike these methods which duplicate full FFNs, MIDUS uniquely decouples capacity from dense computation via retrieval.

---

### Meta-Review · Area_Chair_qhXf · 2025-12-21

**Summary:**

This paper proposes MIDUS, a memory-based alternative to depth up-scaling that replaces duplicated FFN layers with head-wise memory blocks. Reviewers generally found the idea interesting and the implementation careful, and several appreciated the clarity of the design and the efficiency-oriented motivation. At the same time, there was consistent concern about whether the work establishes a strong enough case that memory-based depth up-scaling is a clear improvement over existing approaches, especially closely related sparse or dynamic depth methods.

The rebuttal and revision were thorough and addressed many technical questions, but some reviewers’ core reservations about positioning and scope remain. Given the mixed reviews and remaining concerns, I do not recommend acceptance.

**Reviewer Concerns:**

**1. Concerns largely addressed in the rebuttal**

***(1) Scale and task coverage.***
The authors added 8B-scale experiments and math-reasoning benchmarks, which address the original concern that conclusions were based only on 1B models and knowledge-centric tasks.

***(2) Efficiency and latency reporting.***
The clarification of prefill vs. decoding time, additional scaling plots, and discussion of kernel-level optimizations make the efficiency claims much clearer and more convincing than in the original submission.

***(3) Hyperparameter sensitivity and training fairness.***
The added ablations and explicit statement that training protocols are shared across baselines address concerns about tuning and comparability.


**2. Concerns that remain**

***(1) Missing empirical comparison to closely related methods.***
Several reviewers noted the lack of direct comparisons to recent sparse or dynamic depth approaches (e.g., MoD, DLO). While the authors discuss these methods conceptually, their absence as baselines weakens the empirical positioning of the paper.

***(2) Strength of the central contribution.***
Despite solid engineering and experiments, some reviewers remain unconvinced that replacing FFN duplication with memory retrieval constitutes a sufficiently strong or general advance in scaling methodology, rather than a specific architectural variant with context-dependent benefits.

***(3) Complexity vs. payoff.***
The method introduces additional architectural and implementation complexity (memory banks, retrieval, custom kernels). It is still unclear whether the gains consistently justify this added complexity across settings beyond those studied here.

**Reviewer Scores:**

Based on the rebuttal, I believe:

**Reviewer xNN6** would likely move from reject to borderline, as most factual issues were addressed.

**Reviewer 9Rab** would likely remain borderline.

**Reviewer 9rn7** would likely soften slightly but remain on the reject side, due to missing baselines and positioning concerns.

**Reviewer NZXf** would likely remain positive.

Overall, the post-rebuttal sentiment is still mixed, with no clear consensus in favor of acceptance.

---

### Decision · Program_Chairs · 2026-01-26

Reject